# Local Linearity: the Key for No-regret Reinforcement Learning in Continuous MDPs

**Davide Maran**
Politecnico di Milano, Milan, Italy
davide.maran@polimi.it

**Alberto Maria Metelli**
Politecnico di Milano, Milan, Italy
albertomaria.metelli@polimi.it

**Matteo Papini**
Politecnico di Milano, Milan, Italy
matteo.papini@polimi.it

**Marcello Restelli**
Politecnico di Milano, Milan, Italy
marcello.restelli@polimi.it

## Abstract

Achieving the no-regret property for Reinforcement Learning (RL) problems in continuous state and action-space environments is one of the major open problems in the field. Existing solutions either work under very specific assumptions or achieve bounds that are vacuous in some regimes. Furthermore, many structural assumptions are known to suffer from a provably unavoidable exponential dependence on the time horizon $H$ in the regret, which makes any possible solution unfeasible in practice. In this paper, we identify *local linearity* as the feature that makes Markov Decision Processes (MDPs) both *learnable* (sublinear regret) and *feasible* (regret that is polynomial in $H$). We define a novel MDP representation class, namely *Locally Linearizable MDPs*, generalizing other representation classes like Linear MDPs and MDPS with low inherent Belmman error. Then, i) we introduce CINDERELLA, a no-regret algorithm for this general representation class, and ii) we show that all known learnable and feasible MDP families are representable in this class. We first show that all known feasible MDPs belong to a family that we call *Mildly Smooth MDPs*. Then, we show how any mildly smooth MDP can be represented as a Locally Linearizable MDP by an appropriate choice of representation. This way, CINDERELLA is shown to achieve state-of-the-art regret bounds for all previously known (and some new) continuous MDPs for which RL is learnable and feasible.

## 1 Introduction

*Reinforcement learning* (RL) [35] is a paradigm of artificial intelligence in which an agent interacts with an environment, which is typically assumed to be a Markov Decision Process (MDP) [30], to maximize a reward signal in the long term. By interacting with the environment, an RL algorithm tries to make the agent play actions leading to the highest possible expected reward; RL theory is the field that designs algorithms to be provably efficient, i.e., to work with probability close to one. This idea is formalized in a performance metric called the *(cumulative) regret*, which measures the cumulative difference between actions played by the algorithm and the optimal ones in terms of expected reward.

For the case of episodic *tabular* MDPs, when both the state and the action space are finite, an optimal result was first proved by [4], who showed a bound on the regret of order $\widetilde{\mathcal{O}}(\sqrt{H^3|\mathcal{S}||\mathcal{A}|K})$, where $\mathcal{S}$ is the state space, $\mathcal{A}$ is the action space, $K$ is the number of episodes of interaction, and $H$ the time horizon of every episode. This regret is minimax-optimal in the sense that no algorithm can achieve smaller regret for every arbitrary tabular MDP. This result is not useful in many real-world scenarios, where $\mathcal{S}$ and $\mathcal{A}$ are huge, or even continuous [20, 19, 15]. In fact, all applications of RL

to the physical world, like robotics [20] and autonomous driving [19], have to deal with continuous state spaces. Furthermore, the most common benchmarks used to evaluate practical RL algorithms [36, 7] have continuous state spaces.

One of the first studied families of MDPs with continuous spaces is the Linear Quadratic Regulator (LQR) [5], which goes back to control theory. LQR is a model where the state of the system evolves according to linear dynamics, and the reward is quadratic. Regret guarantees of order $\widetilde{\mathcal{O}}(\sqrt{K})$ for this problem were obtained by [2] (with a computationally inefficient algorithm) and, then, by [12, 9]. Still, this parametric model of the environment is very restrictive and does not capture the vast majority of continuous MDPs. A much wider and non-parametric family is given by *Lipschitz MDPs* [31], which assume that bounded differences in the state-action pair $(s, a)$ correspond to bounded differences in the reward $r(s, a)$ and in the transition function $p(\cdot|s, a)$ (e.g., in Wasserstein metric). Lipschitz MDPs have been applied to several scenarios, like policy gradient methods [29, 3, 28], RL with delayed feedback [22], configurable RL [27], and auxiliary tasks for imitation learning [11, 26]. While this model is very general, its regret guarantees are weak, both in terms of dependence on $K$ and on $H$. In fact, very recently, [25] showed a regret lower bound of order $\Omega(2^H K^{\frac{d+1}{d+2}})$, where $d$ is the dimension of the state-action space, which makes this family of problems *statistically unfeasible*.

Another part of the literature has focused instead on *representation classes* of MDPs. In this paper, we call "representation class" a family of problems that depend both on an MDP and on an exogenous element, usually a *feature map*. One example can be found in the popular class of *Linear MDPs* [40, 18], which assumes that both the transition and the reward function can be factorized as a scalar product of a known feature map $\phi$ of dimension $d_\phi$ and some unknown vectors. Regret bounds of order $\widetilde{\mathcal{O}}(H^2 d_\phi^{3/2} \sqrt{K})$ are possible [18], which succeed in moving the complexity of the problem into the dimension $d_\phi$ of the feature map. Unfortunately, this representation class is very restrictive: i) linearity is assumed on both the $p_h$ and $r_h$ functions; ii) the same linear factorization must be constant along the state-action space $\mathcal{S} \times \mathcal{A}$, which goes in the opposite direction w.r.t. the locality principle introduced by Lipschitz MDPs. The first issue is solved by [42], which significantly extends this class of process by assuming linearity on the Bellman optimality operator, which turns out to be much weaker. This representation class is known as MDPs with *low inherent Bellman error*, a generalization of linear MDPs that further allows for a small approximation error $\mathcal{I}$.

Very recently, different kinds of assumptions for continuous spaces were introduced. In *Kernelized MDPs* [41], both the reward function and the transition function belong to a *reproducing kernel Hilbert space* (RKHS) induced by a known kernel coming from the Matérn covariance function with parameter $\nu > 0$. The higher the value of $\nu$, the more stringent the assumption, as the corresponding RHKS contains fewer functions. This kind of assumption enforces the smoothness of the process, which is stronger for higher values of $\nu$. A generalization of this family can be found in *Strongly smooth MDPs* [25], which require the transition and reward functions to be $\nu-$times continuously differentiable. Although it is a wide, non-parametric family of processes, enforcing the smoothness of the transition function is rather demanding as it implies that the state $s'$ is affected by a smooth noise. For this reason, [25, 24] also defines the larger family of *Weakly smooth MDPs*, which only requires the smoothness of the Bellman optimality operator. This model is extremely general, as it can also capture Lipschitz MDPs. Still, for the same reason, it is affected by an exponential lower bound in $H$. Lastly, note that for the last three kinds of MDPs, *Kernelized*, *Strongly and Weakly Smooth*, the best-known regret bounds are linear in $K$ for some values of $d$ and $\nu$. As the regret is trivially bounded by $K$, these bounds are vacuous, not guaranteeing convergence to the performance of the optimal policy.

**Our contributions.** In this paper, we argue that the aspect that makes many continuous RL problems both learnable and feasible is local linearity, i.e., the possibility to locally approximate the MDP as a process exhibiting some sort of linearity for a well-designed feature map. To support this thesis, in the first part of the paper, $(i)$ we introduce *Locally Linearizable MDPs*, a novel *representation* class of MDPs depending on both a feature map $\phi_h$ and a partition $\mathcal{U}_h$ of the state-action space of the MDP. This class generalizes both LinearMDPs and low inherent Bellman Error while also allowing the feature map to be local. $(ii)$ We design an algorithm, CINDERELLA, which enjoys satisfactory regret bounds for this representation class. In the second part of the paper, we explore how this approach can be compared to the state of the art on continuous MDP; $(iii)$ we show that all families all families that were defined in the continuous RL literature, to the best of our knowledge, for which learnable and feasible RL is possible are included in a novel family of *Mildly Smooth MDPs* defined in this

paper; (*iv*) we show that this family is, in turn, a special instance of our *Locally Linearizable MDPs* representation class, for an appropriate choice of the feature map. Therefore, CINDERELLA can be applied to learning on all these families of continuous MDPs. (*v*) we finally prove that the regret bound of CINDERELLA outperforms several state-of-the-art results in this field.

## 2 Background and set-up

**Markov Decision processes.** We consider a finite-horizon Markov decision process (MDP) [30] $M = (\mathcal{S}, \mathcal{A}, p, r, H)$, where $\mathcal{S}$ is the state space, $\mathcal{A}$ is the action space,[1] $p = \{p_h\}_{h=1}^{H-1}$ is the sequence of transition functions mapping, for each step $h \in [H-1] := \{1, \ldots, H-1\}$, a pair $z = (s, a) \in \mathcal{Z}$ to a probability distribution $p_h(\cdot|z)$ over $\mathcal{S}$, while the initial state $s_1$ may be arbitrarily chosen by the environment at each episode; $r = \{r_h\}_{h=1}^{H}$ is the sequence of reward functions, mapping, for each step $h \in [H]$, a pair $z = (s, a)$ to a real number $r_h(z)$, and $H$ is the horizon. At each episode $k \in [K]$, the agent chooses a policy $\pi_k = \{\pi_{k,h}\}_{h=1}^{H}$, which is a sequence of step-dependent mappings from $\mathcal{S}$ to probability distributions over $\mathcal{A}$. For each step $h \in [H]$, the action is chosen as $a_h \sim \pi_{k,h}(\cdot|s_h)$ and the agent gains reward of mean $r_h(z_h)$ and independent on the past, then the environment transitions to the next state $s_{h+1} \sim p_h(\cdot|z_h)$. For a summary of this notation see A.

**Value functions and Bellman operators.** The state-action value function (or *Q-function*) quantifies the expected sum of the rewards obtained under a policy $\pi$, starting from a state-step pair $(s, h) \in \mathcal{S} \times [H]$ and fixing the first action to some $a \in \mathcal{A}$:

$$Q_h^\pi(s, a) := \mathbb{E}_\pi\left[\sum_{\ell=h}^{H} r_\ell(s_\ell, a_\ell)\bigg| s_h = s, a_h = a\right] = \mathbb{E}_\pi\left[\sum_{\ell=h}^{H} r_\ell(z_\ell)\bigg| z_h = (s, a)\right], \quad (1)$$

where $\mathbb{E}_\pi$ denotes expectation w.r.t. to the stochastic process $a_h \sim \pi_h(\cdot|s_h)$ and $s_{h+1} \sim p_h(\cdot|z_h)$ for all $h \in [H]$. The state value function (or *V-function*) is defined as $V_h^\pi(s) := \mathbb{E}_{a \sim \pi_h(\cdot|s)}[Q_h^\pi(s, a)]$, for all $s \in \mathcal{S}$. The supremum of the value functions across all the policies is referred to as the optimal value function: $Q_h^\star(z) := \sup_\pi Q_h^\pi(z)$ for the Q-function and $V_h^\star(s) := \sup_\pi V_h^\pi(s)$ for the V-function.

In this work, as often done in the literature [43], we make the following

**Assumption 1.** *The instantaneous reward minus its mean $r_h(s, a)$ is $1-$subgaussian, and normalized so that $0 \le Q_h^\pi(z) \le 1$ for every policy $\pi$, $z \in \mathcal{Z}$ and $h \in [H]$.*

Passing to the case where the *per-step* reward is in $[0, 1]$ requires multiplying all upper bounds by $H$. An explicit way to find the optimal value function is given by the *Bellman optimality operator*, which is defined, for every function $f : \mathcal{Z} \to \mathbb{R}$, as $\mathcal{T}_h f(s, a) := r_h(s, a) + \mathbb{E}_{s' \sim p_h(\cdot|s, a)}\left[\sup_{a' \in \mathcal{A}} f(s', a')\right]$. In fact, it is easy to show that $Q_h^\star = \mathcal{T}_h Q_{h+1}^\star$ at every step, while the optimal state-value function is obtained simply as $V_h^\star(a) = \sup_{a \in \mathcal{A}} Q_h^\star(s, a)$.[2]

**Agent's regret.** We evaluate the performance of an agent, i.e., of a policy $\pi_k$ played at episode $k \in [K]$, with its expected total reward, i.e., the V-function evaluated in the initial state $V_1^{\pi_k}(s_1^k)$. The goal of the agent is to play a sequence of policies $\{\pi_k\}_{k=1}^{K}$ to minimize the cumulative difference between the optimal performance $V_1^\star(s_1^k)$ and its performance $V_1^{\pi_k}(s_1^k)$, given the initial state $s_1^k$ chosen by the environment. This quantity takes the name of *(cumulative) regret*, $R_K := \sum_{k=1}^{K} \left(V_1^\star(s_1^k) - V_1^{\pi_k}(s_1^k)\right)$. This quantity is non-negative, and by the normalization condition, we can see that it cannot exceed $K$ as every term in the sum is bounded by $1$. Note that if $R_K = o(K)$, then the average performance of the chosen policies will converge to optimal performance. An algorithm choosing a sequence of policies with this property is called *no-regret*.

**Representation classes of MDPs.** As anticipated in the introduction, we call "representation class" a family of MDPs that is defined through its relation with a feature map or another exogenous element. While the most popular representation class is the Linear MDP, assuming the exact factorization of both $p_h$ and $r_h$, no-regret learning is possible for a much wider family, only requiring a form of *approximate* linearity on the application of Bellman's optimality operator. This class was introduced

---

[1]For convenience, we will denote $\mathcal{Z} = \mathcal{S} \times \mathcal{A}$ and with $z$ any pair $(s, a)$.

[2]The existence of optimal policies is more subtle than in the finite-action case [6], but this does not prevent us from defining a meaningful notion of regret.

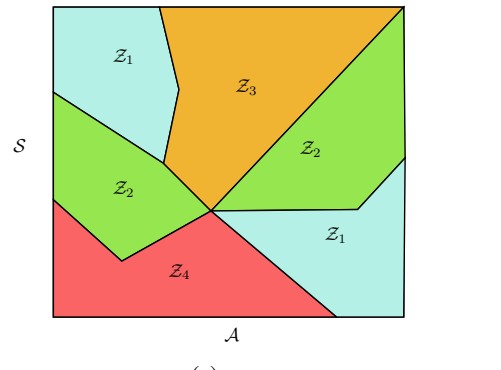
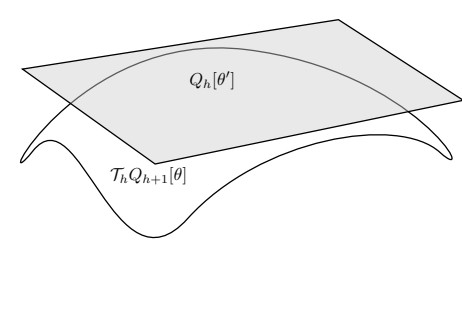

(a)                                                                  (b)

Figure 1: In Locally Linearizable MDPs, we have that, as shown in (a), the space $\mathcal{Z}$ is partitioned into several regions, which do not need to be convex nor connected. On each of these regions, as shown in (b), the result of the Bellman optimality operator can be well approximated by a $Q$ function that is linear in the feature map, with a parameter $\theta$ that may depend on the region itself.

by [42] as MDPs with *low inherent Bellman error*. Given a sequence of compact sets $\mathcal{B}_h \subset \mathbb{R}^{d_\phi}$, and calling $Q_h[\theta](s,a)$ the linear function $\phi_h(s,a)^\top \theta$, the inherent Bellman error w.r.t. $\{\mathcal{B}_h\}_h$ is defined as:

$$\mathcal{I}(\phi) := \max_{h \in [H]} \sup_{\theta \in \mathcal{B}_{h+1}} \inf_{\theta' \in \mathcal{B}_h} \|\phi_h(\cdot)^\top \theta' - \mathcal{T}_h Q_{h+1}[\theta](\cdot)\|_{L^\infty}, \tag{2}$$

where the supremum norm $\|\cdot\|_{L^\infty}$ indicates the maximum of the function in absolute value over $\mathcal{Z}$. In the realizable case (i.e., $\mathcal{I} = 0$), these processes are a strict generalization of Linear MDPs [42]. To achieve regret guarantees for continuous state-action MDPs that go beyond this linear case, we will need to borrow some concepts of smoothness from mathematical analysis, as presented below.

**Smooth functions.** Let $\Omega \subset [-1,1]^d$ and $f : \Omega \to \mathbb{R}$. We define a multi-index $\boldsymbol{\alpha}$ as a tuple of non-negative integers $(\alpha_1, \dots \alpha_d)$. We say that $f \in \mathcal{C}^\nu(\Omega)$, for $\nu \in (0, +\infty)$, if it is $\nu_*$−times continuously differentiable for $\nu_* := \lceil \nu - 1 \rceil$, and there exists a constant $L_\nu(f)$ such that:

$$\forall \boldsymbol{\alpha} : \|\boldsymbol{\alpha}\|_1 = \nu_*, \qquad \forall x,y \in \Omega : \quad |D^{\boldsymbol{\alpha}} f(x) - D^{\boldsymbol{\alpha}} f(y)| \leq L_\nu(f)\|x-y\|_\infty^{\nu - \nu_*} \tag{3}$$

the multi-index derivative is defined as $D^{\boldsymbol{\alpha}} f := \frac{\partial^{\alpha_1 + \dots + \alpha_d}}{\partial x_1^{\alpha_1} \dots \partial x_d^{\alpha_d}}$. The previous set becomes a normed space when endowed with a norm $\|f\|_{\mathcal{C}^\nu}$ defined as $\max \left\{ \max_{|\boldsymbol{\alpha}| \leq \nu_*} \|D^{\boldsymbol{\alpha}} f\|_{L^\infty}, L_\nu(f) \right\}$. Note that, when $\nu \in \mathbb{N}$, this norm reduces to $\|f\|_{\mathcal{C}^\nu} = \max_{|\boldsymbol{\alpha}| \leq \nu} \|D^{\boldsymbol{\alpha}} f\|_{L^\infty}$, since the Lipschitz constant $L_\nu(f)$ of the derivatives up to order $\nu_* = \nu - 1$ correspond exactly to the upper bound of the derivatives of order $\nu$ (which exists as a Lipschitz function is differentiable almost everywhere). For these values of $\nu$, the spaces defined here are equivalent to the spaces $\mathcal{C}^{\nu-1,1}(\Omega)$ defined in [25].

## 3   Locally Linearizable MDPs

As we have stated in the introduction, the main limitation of the low inherent Bellman error assumption is that it cannot model scenarios where the linear parameter $\theta$ changes across the state-action space. In fact, $\phi_h(s,a)^\top \theta$ must be an approximation of the Q-function $Q_h(s,a)$ *uniformly* over $(s,a) \in \mathcal{Z}$. To overcome this limitation, we introduce a novel concept of *locality* to enable the feature map to be associated with different parameters $\theta$ in different regions of the state-action space.

Therefore, from this point on, we are going to associate to a given Markov Decision Process $M$ the following two entities:

1. $\phi_h : \mathcal{Z} \to \mathbb{R}^{d_h}$: a feature map which is allowed to depend on the current stage (also for its dimension.

2. $\mathcal{U}_h$: a sequence of partitions of the state-action space $\mathcal{Z}$ in $N_h$ regions, so that $\mathcal{U}_h :=$ $\{\mathcal{Z}_{h,n}\}_{n=1}^{N_h}$. We call $\rho_h : \mathcal{Z} \to [N_h]$ the map linking every element $z \in \mathcal{Z}$ to the index of its set in the partition $\mathcal{U}_h$.

These two elements are necessary to introduce the function class we are going to use as function approximator in this setting. Calling $\boldsymbol{\theta}_h = \{\theta_{h,n}\}_{n=1}^{N_h}$ a list of vectors in $\mathbb{R}^{d_h}$, one for each of the regions $\mathcal{Z}_{h,n}$, we employ, as function approximator for the state-action value function, the following set

$$\mathcal{Q}_h := \left\{ Q_h[\boldsymbol{\theta}_h](\cdot) = \phi_h(\cdot)^\top \theta_{h,\rho_h(\cdot)}, \quad \text{where} \quad \boldsymbol{\theta}_h = \{\theta_{h,n}\}_{n=1}^{N_h}, \ \theta_{h,n} \in \mathcal{B}_{h,n} \right\}. \tag{4}$$

Coherently, we will call $\mathcal{V}_h := \{V(s) = \sup_{a \in \mathcal{A}} Q(s,a) : Q \in \mathcal{Q}_h, s \in \mathcal{S}\}$. For fixed $z$, we have $Q_h[\boldsymbol{\theta}_h](z) = \phi_h(z)^\top \theta_{h,\rho_h(z)}$, so that the feature map is allowed to depend directly on $z$, while the linear parameter only depends on $\rho_h(z)$, the function indicating in which of the $N_h$ regions we are. $\mathcal{B}_{h,n}$ are arbitrary compact sets which contain the candidate values for the linear parameters $\theta_{h,n}$. Two relevant quantities for this model are the $2-$norm of the feature map and the diameter of the sets $\mathcal{B}_{h,n}$,

$$L_\phi := \sup_{h \in [H], z \in \mathcal{Z}} \|\phi_h(z)\|_2 \qquad \mathcal{R}_{h,n} := \text{diam}(\mathcal{B}_{h,n}),$$

Respectively. Analog normalization constants appear in [18, 42]. The low inherent Bellman error property can be redefined in this setting as follows.

**Definition 1.** *(Inherent Bellmann Error) Given a family of compact sets $\mathcal{B}_{h,n} \subset \mathbb{R}^{d_h}$ depending on $h \in [H], n \in [N_h]$, and their Cartesian product $\mathcal{B}_h = \bigtimes_{n=1}^{N_h} \mathcal{B}_{h,n}$, we define:*

$$\mathcal{I}(\phi, \mathcal{U}) := \max_{h \in [H]} \sup_{\boldsymbol{\theta}_{h+1} \in \mathcal{B}_{h+1}} \inf_{\boldsymbol{\theta}_h \in \mathcal{B}_h} \|Q_h[\boldsymbol{\theta}_h](\cdot) - \mathcal{T}_h Q_{h+1}[\boldsymbol{\theta}_{h+1}](\cdot)\|_{L^\infty}. \tag{5}$$

Note that, for $N_h = 1$, our definition exactly reduces to Equation (2), as expected. As in that case, the term $\mathcal{I}$ plays the role of an approximation error. By assuming a bound on $\mathcal{I}(\phi, \mathcal{U})$, we can now give a formal definition of the class of MDPs we are going to study in this paper.

**Definition 2.** *An $\mathcal{I}-$Locally Linearizable MDP[3] is a triple $(M, \{\phi_h\}_{h=1}^H, \{\mathcal{U}_h\}_{h=1}^H)$ where $M$ is an MDP, $\phi_h : \mathcal{Z} \to \mathbb{R}^{d_h}$ is a feature map and $\mathcal{U}_h$ a sequence of partitions of the state-action space $\mathcal{Z}$ in $N_h$ regions such that the corresponding inherent Bellman error (definition 1) satisfies $\mathcal{I}(\phi, \mathcal{U}) \leq \mathcal{I}$.*

As for the class of MDPs with Low Inherent Bellmann error, $\phi$ (and $\mathcal{U}$ in our case) must be known to the agent, while $\mathcal{I}$ is not needed. The novel aspect this class is that, as a result of definition 1, linearity is independently enforced in separate regions of the state-action space. We provide a visualization of this concept in Figure 1. Note that, in principle, any MDP belongs to the Locally Linearizable representation class for $\mathcal{I} = 1$. In fact, if we take $\mathcal{U}_h = \{\mathcal{Z}\}$, the trivial partition containing just the state-action space as an element, and $\phi_h(z) = 0$ (a feature map mapping everything to 0), Equation (5) is satisfied with $\mathcal{I} = 1$. Nonetheless, this class is *interesting* only if $\mathcal{I}$ is small, as it is easy to show that the regret of any algorithm in this class grows at least as $\mathcal{I}K$.

**Limitations of known approaches.** Formally, no algorithm in theoretical RL literature can achieve no-regret learning on this class of problems, as it is a superclass of the low inherent Bellman error, which, to the best of our knowledge, is not included in any other setting that has been tackled. Still, a clever strategy called feature extension (example 2.1 from [18]) may allow us to solve this class of MDPs. In fact, consider ELEANOR [42], the only algorithm able to deal with MDPs with low inherent Bellman error. This trick employs a newly defined feature map:

$$\widetilde{\phi}_h(z) := \underbrace{[\delta_{h,1}(z)\phi_h(z), \ \delta_{h,2}(z)\phi_h(z), \ldots, \delta_{h,N_h}(z)\phi_h(z)]^\top}_{N_h}, \quad \delta_{h,n}(z) := \begin{cases} 1 & \text{if } \rho_h(z) = n \\ 0 & \text{otherwise} \end{cases}$$

so that its dimension expands from $d_h$ to $N_h d_h$. This way, any function of $\mathcal{Q}_h$ (as defined in Equation 4) is linear in $\widetilde{\phi}_h(z)$, with a single $\theta_h$ independent of the region $\mathcal{Z}_{n,h}$. Indeed, we have for $Q_h[\boldsymbol{\theta}_h] \in \mathcal{Q}_h$:

---

[3]For readability, the bound $\mathcal{I}$ on the inherent Bellman error will be dropped when talking about this class in general.

**Algorithm 1** CINDERELLA (Constrained INDEpendent REgressions with Local Linear Approximations)

---

**Require:** Failure probability $\delta$, Regularization $\lambda$, Region mapping $\rho_h$, Feature mapping $\phi_h$
1: Initialize $\Lambda_{h,n}^1 := \lambda I$ for every $h \in [H], n \in [N_h]$
2: **for** $k = 1, 2, \ldots K$ **do**
3:     Receive initial state $s_1^k$
4:     Solve optimization program in (7) obtaining $\overline{\theta}_{h,n}^k$ for every $h \in [H]$ and $n \in [N_h]$
5:     $Q_h[\overline{\boldsymbol{\theta}}_h^k](z) = \phi_h(z)^\top \overline{\theta}_{h,\rho_h(z)}^k$  $\forall h \in [H]$
6:     **for** $h = 1, 2, \ldots H - 1$ **do**
7:         Choose action $a_h^k \in \arg\max_{a \in \mathcal{A}} Q_h[\overline{\boldsymbol{\theta}}_h^k](s_h^k, a)$
8:         Receive reward $r_h^k$ and next state $s_{h+1}^k$
9:     **end for**
10: **end for**

---

$$Q_h[\boldsymbol{\theta}_h](z) = \phi_h(z)^\top \theta_{h,\rho_h(z)} = \widetilde{\phi}_h(z)^\top [\theta_{h,1}, \theta_{h,2}, \ldots, \theta_{h,N_h}].$$

This shows every Locally Linearizable MDP with feature map $\phi_h$ of dimension $d_h$ is also an MDP with low inherent Bellman error w.r.t. $\widetilde{\phi}_h$ of dimension $N_h d_h$ and the same value for $\mathcal{I}$. If we apply the regret bound for ELEANOR [42, Theorem 1], we obtain:

$$R_K = \widetilde{\mathcal{O}}\left( \sum_{h=1}^{H} N_h d_h \sqrt{K} + \sum_{h=1}^{H} \sqrt{N_h d_h} \mathcal{I} K \right), \tag{6}$$

holding with high probability. The issue is that the second term, growing linearly in $K$, depends on the number of regions as $\sqrt{N_h}$. As we will see in the second part of this paper, the application of this model to Smooth MDPs requires $N_h$ to be very large and also dependent on $K$. For this reason, in the next section, we introduce an *ad hoc* algorithm for Locally Linearizable MDPs to improve this dependence.

### 3.1 Algorithm

As we have seen, even a wise application of the ELEANOR algorithm is not enough to solve our setting of Locally Linearizable MDPs satisfyingly. Thus, we introduce a novel algorithm, CINDERELLA (Algorithm 1). Before analyzing it, we need to introduce some notation. Let us call $s_h^k, a_h^k, r_h^k$ the state, action, and reward relative to step $h$ of episode $k$. Moreover, we denote $z_h^k := (s_h^k, a_h^k)$ and $\phi_h^k = \phi_h(z_h^k)$. At every episode $k \in [K]$, we compute an optimistic estimation of the $Q$-function for every step $h \in [H]$. Then, we choose actions in order to maximize this function while fixing $s_1^k$ (line 7). Clearly, what is really important is how this surrogate $Q$-function is computed (line 4). Here, CINDERELLA relies on solving an optimization problem (7), which follows an idea similar to the one of [42]. We want to optimize over three sets of variables: $\widehat{\theta}_{h,n}, \overline{\xi}_{h,n}, \overline{\theta}_{h,n}$, the first one representing ridge regression of the linear parameter for region $n$ at step $h$, the second representing the uncertainty relative to this estimation, and the third one an "optimistic" estimate. Under this view, the objective is to maximize the surrogate $V$-function in the first state, and the constraints are designed so that all variables match their intuitive interpretation. Formally, the optimization problem is defined as:

$$\max_{\widehat{\theta}_{h,n}, \overline{\xi}_{h,n}, \overline{\theta}_{h,n}} \max_{a \in \mathcal{A}} \phi_1(s_1^k, a)^\top \overline{\theta}_{1, \rho_1(s_1^k, a)} \tag{7}$$

$$\text{s.t.} \quad \widehat{\theta}_{h,n} = {\Lambda_{h,n}^k}^{-1} \sum_{\tau=1}^{k-1} \mathbf{1}\{\rho_h(z_h^\tau) = n\} \phi_h^\tau \left( r_h^\tau + \max_{a \in \mathcal{A}} \phi_{h+1}(s_{h+1}^\tau, a)^\top \overline{\theta}_{h+1, \rho(s_{h+1}^\tau, a)} \right)$$

$$\overline{\theta}_{h,n} = \widehat{\theta}_{h,n} + \overline{\xi}_{h,n}$$

$$\|\overline{\xi}_{h,n}\|_{\Lambda_{h,n}^k} \leq \sqrt{\alpha_{h,n}^k}.$$

Where $\Lambda_{h,n}^k := \sum_{\tau=1}^{k} \mathbf{1}\{\rho_h(z_h^\tau) = n\} \phi_h^\tau {\phi_h^\tau}^\top + \lambda I$, is the design matrix of $\lambda-$regularized ridge regression, and $\alpha_{h,n}^k$ is a constant determining the exploration rate which will be fixed in the analysis B.6. As it is for ELEANOR, this algorithm turns out to be computationally inefficient, an issue that we

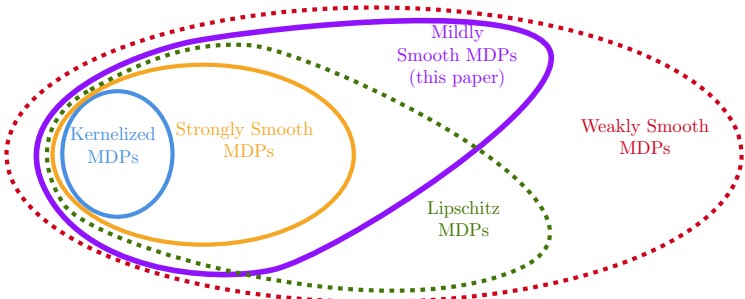

Figure 2: Relation between the setting described in this paper and the other settings proposed for reinforcement learning in continuous state-action spaces. The dashed line means that inclusion holds, but passing to the larger family brings a $\exp(H)$ lower bound on the regret. As we can see, the Mildly smooth MDP is the largest known setting for which regret of order poly(H) is possible. Note that the Strongly Smooth family also contains known families like LQRs and Linear MDPs with smooth feature map [25].

discuss in the Appendix D. The first constraint enforces that $\widehat{\theta}_{h,n}$ is estimated with ridge regression having as target the (optimistic) value function estimated for step $h+1$, and the second constrain $\overline{\theta}_{h,n} = \widehat{\theta}_{h,n} + \overline{\xi}_{h,n}$ ensures that the optimistic estimate in every region is given by its mean estimate plus the uncertainty, while the third one bounds the magnitude of $\overline{\xi}_{h,n}$ so that this uncertainty shrinks the more data we collect.

Although the structure of the algorithm is directly inherited from ELEANOR, CINDERELLA distinguishes itself by dividing the samples across the various regions of the state-action space of the MDP. This allows parameters associated with different regions to be learned independently. Despite introducing another layer of technical difficulty in proving the regret bound, this procedure is relatively natural given the characteristics of our class of problems.

**CINDERELLA: Regret bound.** Having defined the algorithm, we can state a theorem showing a high probability regret bound in our setting.

**Theorem 2.** *Assume to be in an $\mathcal{I}-$Locally Linearizable MDP with $L_\phi = \mathcal{O}(1)$, $\sup_{n \in [N_h]} \mathcal{R}_{h,n} = \mathcal{O}(\sqrt{d_h})$ and that Assumption 1 holds. Then, with probability at least $1 - \delta$, CINDERELLA (Algorithm 1), with $\lambda = 1$ achieves a regret bound of order*

$$R_K = \widetilde{\mathcal{O}}\left(\sum_{h=1}^{H} N_h d_h \sqrt{K} + \sum_{h=1}^{H} \sqrt{d_h} \mathcal{I} K\right).$$

The proof of this result is long, and is deferred to sections B and C of the appendix. Note that the two normalization assumptions $L_\phi = \mathcal{O}(1)$ and $\sup_{h \in [H], n \in [N_h]} \mathcal{R}_{h,n} = \mathcal{O}(\sqrt{d_h})$ correspond to the ones enforced by [42]. While for $N_h = 1$ the two reported bounds coincide, Theorem 2 proves superior to the regret bound obtained for ELEANOR in our setting (Equation 6), as it prevents the second term, which is linear in $K$, to depend on $N_h$. This dramatically changes the potential of the regret bounds and, as we will see in the next sections, represents the key to achieving sub-linear regret bounds for RL in continuous state-action spaces.

## 4 From local linearity to Mildly Smooth MDPs

Having proved that our CINDERELLA algorithm enjoys an improved regret bound on Locally Linearizable MDPs, we need to see how powerful this new class is once applied to problems with continuous state-action spaces for which a representation is not given a priori. To this aim, we are going to define a family of continuous-space MDPs, and prove that they are included in the Locally Linearizable class for a particular choice of $\phi_h, \mathcal{U}_h$. From now on, we assume, without loss of generality, that $\mathcal{S} = [-1, 1]^{d_S}$ and $\mathcal{A} = [-1, 1]^{d_A}$, so that $\mathcal{Z} = [-1, 1]^d$. We call Mildly Smooth MDP a process where the Bellman optimality operator outputs functions that are smooth.

**Definition 3.** *(Mildly Smooth MDP). An MDPs is* Mildly Smooth *of order $\nu$ if, for every $h \in [H]$, the Bellman optimality operator $\mathcal{T}_h$ is bounded on $L^\infty(\mathcal{Z}) \to \mathcal{C}^\nu(\mathcal{Z})$.*

Boundedness over $L^\infty(\mathcal{Z}) \to \mathcal{C}^\nu(\mathcal{Z})$ means that the operator transforms functions that are bounded (i.e., belong to $L^\infty(\mathcal{Z})$) into functions that are $\nu$-times differentiable (i.e., belong to $\mathcal{C}^\nu(\mathcal{Z})$). Moreover, there exists a constant $C_{\mathcal{T}} < +\infty$ such that $\|\mathcal{T}_h f\|_{\mathcal{C}^\nu} \le C_{\mathcal{T}}(\|f\|_{L^\infty} + 1)$ for every $h \in [H]$ and every function $f \in L^\infty(\mathcal{Z})$. Intuitively, this condition means that by applying the Bellman operator to bounded (possibly non-smooth) functions, we always get functions that are smooth.

In order to reduce this family of processes to Locally Linearizable MDPs, we have to design the partition of the state-action space into sets $\mathcal{Z}_{h,n}$. Since $\mathcal{Z} \subset [-1,1]^d$, we can find, for every $\varepsilon > 0$, a set $\mathcal{Z}^\varepsilon$ which is an $\varepsilon$-cover of $\mathcal{Z}$ in the infinity norm, such that $|\mathcal{Z}^\varepsilon| =: N \le (2/\varepsilon)^d$. Now, for every $z^n \in \mathcal{Z}^\varepsilon$, we define recursively $\mathcal{Z}_n$ to be the set of points which are near to $z^n$, formally:

$$\mathcal{Z}_1 := \{z \in \mathcal{Z} : \|z - z^1\|_\infty \le \varepsilon\}, \qquad \mathcal{Z}_n := \{z \in \mathcal{Z} : \|z - z^n\|_\infty \le \varepsilon\} \setminus \cup_{\ell=1}^{n-1} \mathcal{Z}_\ell. \qquad (8)$$

By definition, every point of $\mathcal{Z}$ is matched with a point $z^n$ of the cover $\mathcal{Z}^\varepsilon$ and, importantly, is assigned to exactly one subset $\mathcal{Z}_n$ of $\mathcal{Z}$. This way, we have defined a partition $\{\mathcal{Z}_n\}_{n=1}^N$ of $\mathcal{Z}$, one that does not depend on the step $h$. Secondly, we have to choose the feature map $\phi_h(\cdot)$. To this end, we define $\phi_h(z)$ as the vector of *Taylor polynomials of degree $\nu_*$* centered in $z^n \in \mathcal{Z}^\varepsilon$ for $n = \rho(z)$ (just "Taylor feature map" in the following). This means that, fixed $z \in \mathcal{Z}$ such that $\rho(z) = n$, the map $\phi(z)$ will contain terms of the form $(z - z^n)^{\boldsymbol{\alpha}}$ for every multi-index $\|\boldsymbol{\alpha}\|_1 \le \nu_*$. Note that the feature map does not depend on the time-step $h$ either. The dimension of this feature map, which we call $d_{\nu_*}$, corresponds to the number of non-negative multi-indexes such that $\|\boldsymbol{\alpha}\|_1 \le \nu_*$, which is well-known to be $d_{\nu_*} = \binom{\nu_* + d}{\nu_*} \le \nu_*^d$. The power of this choice lies in the fact that, as it is well known from mathematical analysis, any $\mathcal{C}^\nu$ function can be approximated by a Taylor polynomial of degree $\nu_*$ in a neighborhood of diameter $\varepsilon$ with an error of order $\varepsilon^\nu$. Seeing the regions $\mathcal{Z}_n$ we have defined as neighborhoods of the points $z^n$ of the cover, the analogy is complete. This argument is made formal in the following result.

**Theorem 3.** *Let $M$ be a Mildly Smooth MDP, and $\varepsilon \le 1/(2C_{\mathcal{T}}H)^{1/\nu}$. Then, choosing $\mathcal{Z}_{h,n}$ as in Equation (8) and taking $\phi_h$ as the Taylor feature map on the same regions, the tuple $(M, \{\mathcal{Z}_{h,n}\}_{n,h}, \{\phi_h\}_h)$ is an $\mathcal{I}-$Locally Linearizable MDP with*

$$L_\phi = 1 + 2\sqrt{d}_{\nu_*}, \qquad \mathcal{R}_{h,n} = 2\sqrt{d}_{\nu_*}C_{\mathcal{T}}, \qquad \mathcal{I} \le 2C_{\mathcal{T}}\varepsilon^\nu.$$

This reduction shows how general the class of Locally Linearizable MDPs is and enables us to tackle Mildly Smooth MDPs with the CINDERELLA algorithm, originally designed for Locally Linearizable MDPs. By appropriately selecting the parameter $\varepsilon$, we can prove the following theorem, bounding the regret of a Locally Linearizable MDP with smoothness $\nu$ and state-action space of dimension $d$.

**Theorem 4.** *Let $M$ be a Mildly smooth MDP of parameter $\nu > 0$ satisfying Assumption 1. With probability at least $1 - \delta$, CINDERELLA, initialized with $\lambda = 1$, $\mathcal{Z}_{h,n}$ as in Equation (8) and $\phi_h$ given by the Taylor feature map on the same regions, achieves a regret bound of order:*

$$R_K \le \widetilde{\mathcal{O}}\left(Hd_{\nu_*}K^{\frac{\nu+2d}{2\nu+2d}} + H^{\frac{2\nu+2d}{\nu}}\right).$$

Before comparing our result with the state of the art, some comments are due. First note that the exponent of $K$ is $\frac{\nu+2d}{2\nu+2d}$, which is always in $(1/2, 1)$. This means that the no-regret property is achieved *in every regime*. Two elements in this regret bound are undesirable: i) the exponential dependence in $d$ (as $d_{\nu_*} \le \nu_*^d$) and ii) the lower-order term $H^{\frac{2\nu+2d}{\nu}}$, which has an exponent that may be very large, albeit polynomial in $H$. The first issue is discussed in the appendix (Section E.5). We show that even for the much simpler continuous *bandit* problem, lower bounds entail that the problem is not learnable unless $d = o(\log(K))$. Therefore, terms of order $2^d$ can still be seen as $o(K^\alpha)$ for an arbitrarily small $\alpha > 0$. For the second issue, note that the exponent of $H$ in the lower order term is significantly large only if $d \gg \nu$. This regime is known to be very difficult, and in the literature before this paper it was not even possible to achieve the no-regret property (even just for $d > 2\nu$).

We end this section with a simple corollary showing how to deal with the "large $\nu$" regime.

**Corollary 5.** *Under the assumption of theorem 4, for $\nu > \log(K)$ we have*

$$R_K = \widetilde{\mathcal{O}}\left(H\log(K)^d K^{\frac{1}{2}} + H^2\right).$$

| Algorithm | Weakly | Lipschitz | **Mildly** | Strongly | Kernelized |
|---|---|---|---|---|---|
| [25] LEGENDRE-ELEANOR | $K^{\frac{\nu+3d/2}{d+2\nu}}$ | $K^{\frac{1+3d/2}{2+d}}$ | $K^{\frac{\nu+3d/2}{2\nu+d}}$ | $K^{\frac{\nu+3d/2}{2\nu+d}}$ | $K^{\frac{\nu+3d/2}{2\nu+d}}$ |
| [17] GOLF | $K^{\frac{2\nu+3d}{4\nu}}$ | $K^{\frac{2+3d}{4}}$ | $K^{\frac{2\nu+3d}{4\nu}}$ | $K^{\frac{2\nu+3d}{4\nu}}$ | $K^{\frac{2\nu+3d}{4\nu}}$ |
| [34] NET-Q-LEARNING | ✗ | $K^{\frac{1+d}{2+d}}$ | ✗ | $K^{\frac{1+d}{2+d}}$ | $K^{\frac{1+d}{2+d}}$ |
| CINDERELLA (**Ours**) | ✗ | ✗ | $K^{\frac{\nu+2d}{2\nu+2d}}$ | $K^{\frac{\nu+2d}{2\nu+2d}}$ | $K^{\frac{\nu+2d}{2\nu+2d}}$ |
| [25] LEGENDRE-LSVI | ✗ | ✗ | ✗ | $K^{\frac{\nu+2d}{2\nu+d}}$ | $K^{\frac{\nu+2d}{2\nu+d}}$ |
| [41] KOVI | ✗ | ✗ | ✗ | ✗ | $K^{\frac{\nu+3d/2}{2\nu+d}}$ |
| $\exp(H)$ lower bound | **Yes** | **Yes** | No | No | No |

Table 1: Table containing the order w.r.t. $K$ of the regret guarantee of each algorithm for each setting discussed in the paper. Columns correspond to different smoothness assumptions: Weakly and Strongly Smooth MDPs were defined in [25], Lipschitz MDPs in [31], and Kernelized MDPs in [41]. Rows correspond to algorithms with no-regret guarantees for some of the settings. [25, 34, 38] represented the state of the art for Strongly smooth MDPs, Lipschitz MDPs, and Kernelized MDPs, respectively. The last row indicates whether the corresponding setting is feasible or if there exists an $\exp(H)$ lower bound for the regret.

*Proof.* Applying theorem 4 for $\nu$ arbitrarily large does not work, as $d_{\nu_*} \approx \nu^d$ makes the bound vacuous. Still, note that, being $\|\cdot\|_{\mathcal{C}^{\nu'}} \leq \|\cdot\|_{\mathcal{C}^{\nu'}}$ for $\nu' \leq \nu$, under the assumption of the theorem we can take $\nu = \log(K)$. At this point, theorem 4 ensures

$$R_K \leq \widetilde{\mathcal{O}}\left(H\log(K)^d K^{\frac{\log(K)+2d}{2\log(K)+2d}} + H^2\right) = \widetilde{\mathcal{O}}\left(H\log(K)^d K^{\frac{1}{2}} + H^2\right),$$

due to the fact that $K^{\frac{\log(K)+2d}{2\log(K)+2d}} = K^{\frac{1}{2}}K^{\frac{d}{2\log(K)+2d}} = K^{\frac{1}{2}}2^{\frac{d\log(K)}{2\log(K)+2d}} \leq 2^d K^{\frac{1}{2}}$. □

The appearance of $\log(K)^d$ should not scare: this term is necessary even in the simpler case of bandits with squared exponential kernel, as the lower bound in [37] shows.

# 5 Comparison with related works

Regret bounds for continuous MDPs have been an area of intense research in recent years. While many parametric families like LQRs have been shown to achieve $\text{poly}(H)\sqrt{K}$ regret bounds [2], tackling this problem in more general cases has been proved to be very challenging.

Kernelized MDPs [8, 41, 13] are a representation class of processes assuming that the transition function $p_h(\cdot)$ as well as the reward function $r_h(\cdot)$ belong to a Reproducing Kernel Hilbert Space (RKHS) with given kernel $k(\cdot,\cdot)$. Most of the literature deals with the case of the Matérn kernel. The smoothness of this kernel is determined by a parameter $\nu > 0$; by fixing it we can compare this family with the other families of continuous MDPs. The best-known result [41] in this setting only achieves regret $\widetilde{\mathcal{O}}(K^{\frac{\nu+3d/2}{2\nu+d}})$, which is vacuous if $d > 2\nu$. Very recently [38] presented a regret bound of order $\widetilde{\mathcal{O}}(K^{\frac{\nu+d}{2\nu+d}})$, but we were not able to verify the correctness of this result. We discuss a possible subtle issue of the proof in Appendix F. Another family that is based on assuming that $p_h$ and $r_h$ belong to some given functional space is the Strongly Smooth MDP [25]. This family assumes that $p_h(s'|\cdot), r_h(\cdot) \in \mathcal{C}^\nu(\mathcal{Z})$. A subtle difference is that in [25], the smoothness index $\nu$ was restricted to be an integer, while in our case, it is a generic real $\nu > 0$. Regret bounds for this family were shown of order $\widetilde{\mathcal{O}}(K^{\frac{\nu+2d}{2\nu+d}})$, which is vacuous even for $d > \nu$. Since, for the same $\nu$, the Matérn kernel RKHS is a subset of $\mathcal{C}^\nu(\mathcal{Z})$ (see Appendix E.3), Strongly Smooth MDPs are more general than the former family. Further increasing the generality, we have Lipschitz MDPs [31]. This is only a superset of the Strongly Smooth MDPs for $\nu = 1$, as Lipschitz MDPs do not admit higher levels of smoothness. A regret guarantee of order $\widetilde{\mathcal{O}}(K^{\frac{1+d}{2+d}})$, which turns out to be optimal for this setting, was achieved by different algorithms [33, 34, 32, 21] in recent years. Unfortunately, it was recently shown that an *exponential dependence on the horizon $H$* is unavoidable [25]. Therefore, the Lipschitz MDPs, as any of their generalizations, are intrinsically unfeasible.

Increasing the generality even further, we find the family of Weakly Smooth MDPs [25], which imposes the Bellman optimality operator $\mathcal{T}_h$ to be bounded on $\mathcal{C}^\nu(\mathcal{Z}) \to \mathcal{C}^\nu(\mathcal{Z})$. For fixed $\nu$, this assumption generalizes Strongly smooth MDPs (with the same $\nu$), and, for $\nu = 1$, the Lipschitz MDPs. For this reason, also this family is affected by an $\exp(H)$ regret lower bound, while in terms of $K$ its regret has been bounded as $\widetilde{\mathcal{O}}(K^{\frac{\nu+3d/2}{2\nu+d}})$, the same as Kernelized MDPs. Introducing a different notion of dimension, we can also define the family of MDPs with bounded Bellman-Eluder dimension [17]. This family is, in a certain sense, even wider, but admits worse regret bounds [25].

**Key point: locality.** Our approach leverages the concept of locality. We approximate the continuous problem by constructing a feature map that captures information from local neighborhoods. This is a novel approach in the context of Reinforcement Learning (RL), with only [38] employing a remotely similar idea. Nonetheless, the effectiveness of this strategy has been well-established in the field of Continuous Armed Bandits, as demonstrated by [16, 23].

**Comparison with our work.** As the name suggests, the family of Mildly Smooth MDPs occupies an intermediate position between Weakly and Strongly Smooth processes. In Appendix E.2, we formally prove this relation, showing that both inclusions are strict. A graphical representation of this relationship is shown in Figure 2. Note that, even if our family is not the largest that has been studied in the literature, it is indeed the largest known one for which RL is feasible.

We have summarized the comparison between the regret bound of CINDERELLA with state-of-the-art algorithms for the various MDP families in Table 1, where settings (columns) are listed from the most general (Weakly Smooth) to the least (Kernelized). All inclusions are intended to hold for a fixed parameter $\nu$, which is shared by all the MDP families apart from the Lipschitz one, which in some sense has fixed $\nu = 1$. For every column, the best regret guarantee is colored in green, with ✗ meaning that the algorithm (row) is unsuitable for the setting (column). As we can see, CINDERELLA i) works in every setting where there is no $\exp(H)$ lower bound for the regret, achieving the best guarantees ii) is the only algorithm to exploit smoothness fully, i.e. achieving regret $\sqrt{K}$ in the limit $\nu \to \infty$ while being non-vacuous for all finite values of $d, \nu$.

## 6    Conclusion

In this paper, we have significantly enlarged the set of MDPs for which no-regret guarantees are possible. After defining the representation class of Locally Linearizable MDPs (Section 3), we have introduced a new algorithm called CINDERELLA (Algorithm 1), which is able to achieve a strong regret guarantee on this setting, being a "local" generalization of the ELEANOR algorithm. In the second part of the paper, we have introduced a family called Mildly Smooth MDPs (Section 4), which generalizes all the known continuous MDP families where no-regret learning is feasible. Through an argument based on local Taylor polynomial approximation, we have proved that this family is a special instance of the Locally Linearizable MDPs, with a specific partition and feature map. Therefore, we were able to achieve, in Theorem 4, a regret bound for CINDERELLA in the family of Mildly Smooth MDPs, which constitutes the main result of this paper. Not only this bound surpasses the state-of-the-art in terms of generality for feasible settings, but is also able to achieve improved regret bounds for Strongly Smooth MDPs and Kernelized MDPs. Crucially, our work proves the first regret bound that is non-vacuous in any regime for these two families.

## Acknowledgements

Funded by the European Union – Next Generation EU within the project NRPP M4C2, Investment 1.,3 DD. 341 - 15 march 2022 – FAIR – Future Artificial Intelligence Research – Spoke 4 - PE00000013 - D53C22002380006.

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

# A    Table of Notation

| | |
|---|---|
| $\mathcal{S}, \mathcal{A}$ | State/Action space |
| $p_h, r_h$ | transition/reward function at step $h$ |
| $H$ | time horizon |
| $\mathcal{Z}$ | $\mathcal{S} \times \mathcal{A}$ |
| $Q_h^\star, V_h^\star$ | Optimal state-action/state value function at step $h$ |
| $\mathcal{T}_h$ | Bellman optimality operator at step $h$ |
| $\phi_h$ | feature map at step $h$ |
| $d_h$ | dimension of $\phi_h$ |
| $N_h$ | number of regions at step $h$ |
| $L_\phi$ | bound over the two-norm of $\phi_h$ for every $h \in [H]$ |
| $\mathcal{Z}_{h,n}$ | Element of a partition of $\mathcal{Z}$ at step $h$ |
| $\rho_h$ | Mapping from $\mathcal{Z}$ to index in $[N_h]$ |
| $\mathcal{U}_h$ | $\{\mathcal{Z}_{h,n} : u = 1, \dots N_h\}$ |
| $Q_h[\boldsymbol{\theta}_h]$ | Q-function associated to given $\boldsymbol{\theta}_h$ parameter at step $h$. |
| $V_h[\boldsymbol{\theta}_h]$ | $\max_{a \in \mathcal{A}} Q_h[\boldsymbol{\theta}_h](\cdot, a)$ |
| $\mathcal{B}_h$ | Set of candidate $\boldsymbol{\theta}_h$ at step $h$ |
| $\mathcal{B}_{h,n}$ | Set of candidate $\theta_{h,n}$ at step $h$ restricted to the region $n$ |
| $\mathcal{R}_{h,n}$ | Norm-2 radius of $\mathcal{B}_{h,n}$ |
| $\mathcal{R}_{h,\max}$ | $\sup_{n \in [N_h]} \mathcal{R}_{h,n}$ |
| $\mathcal{Q}_h, \mathcal{V}_h$ | Space of candidate state-action/state value functions at step $h$ |
| $\boldsymbol{\theta}_h^\star$ | See equation (10) |
| $s_h^k, a_h^k, r_h^k$ | state/action/reward relative to step $h$ of episode $k$ |
| $z_h^k$ | $(s_h^k, a_h^k)$ |
| $\phi_h^k$ | $\phi_h(z_h^k)$ |
| $\Lambda_{h,n}^k$ | Ridge regression matrix for space $\mathcal{Z}_{h,n}$ up to episode $k$ |
| $\overline{\xi}_{h,n}, \overline{\theta}_{h,n}, \widehat{\theta}_{h,n}$ | Optimization variables of the problem in section B.2 |
| $\overline{Q}_h^k(z)$ | $Q_h[\overline{\boldsymbol{\theta}}_h^k](z)$ |
| $\overline{V}_h^k(s)$ | $\max_{a \in \mathcal{A}} \overline{Q}_h^k(s, a)$ |
| $\eta_h^k(V)$ | Bellman error at step $h$ of episode $k$ w.r.t. $V$ |
| $\overset{\circ}{\theta}_{h,n}(Q_{h+1})$ | see equation 12 |
| $\Delta_h(Q_{h+1})$ | Error done approximating $Q_{h+1}$ with $\overset{\circ}{\boldsymbol{\theta}}_h(Q_{h+1})$ |
| $\rho_{h,n}^k$ | $\mathbf{1}\{\rho_h(z_h^k) = n\}$ |
| $\sqrt{\beta_{h,n}^k}$ | Quantity defined by corollary 10 |
| $\mathcal{N}(\cdot)$ | Covering number in infinity norm |
| $E$ | Good event |

| | |
|---|---|
| $\Omega$ | Generic subset of $[-1, 1]^d$ |
| $d$ | dimension of a set (usually $\mathcal{Z} = [-1, 1]^d$) |
| $\nu$ | Index saying how many times a function is differentiable |
| $\nu_*$ | $\lceil \nu - 1 \rceil$ |
| $L_\nu(f)$ | Lipschitz constant of $f$ w.r.t. the index $\nu$ |
| $\boldsymbol{\alpha}$ | multi-index |
| $D^{\boldsymbol{\alpha}}$ | multi-index derivative |
| $\mathcal{C}^\nu$ | space of functions that are $\nu-$times differentiable |
| $C_{\mathcal{T}}$ | See definition 3 |
| $\|\cdot\|_{L^\infty}$ | supremum norm of a function, $\|f\|_{L^\infty} = \sup|f|$ |
| $\|\cdot\|_{\mathcal{C}^\nu}$ | norm over $\mathcal{C}^\nu$ |
| $T_y^{\nu_*}[f]$ | Taylor polynomial of $f$ of order $\nu_*$ centered in $y$ |
| $d_{\nu_*}$ | $\binom{\nu_* + d}{\nu_*}$ |

## B  Locally Linearizable MDPs

As stated in the main paper, Locally Linearizable MDPs are a representation class of processes that can be efficiently approximated with linear function in some predefined regions of $\mathcal{Z}$.

Precisely, the linearity stays in the iterations of the Bellman optimality operator, which we recall here. For a general MDP, we call $\mathcal{S}, \mathcal{A}$ the state and action space, respectively, $\mathcal{Z} = \mathcal{S} \times \mathcal{A}$, and $\mathcal{T}_h$ the Bellman optimality operator at step $h$, which is given, for every function $f : \mathcal{Z} \to [0, 1]$, by

$$\mathcal{T}_h f(z) := \int_{\mathcal{S}} p_h(s'|z) \max_{a' \in \mathcal{A}} f(s', a') \, ds'.$$

Recalling the definition given in the main paper, in a Locally Linearizable MDP the state-action space $\mathcal{Z}$ is partitioned into a step-dependent number $N_h$ of regions $\mathcal{Z}_{h,n}$ that we collect into $\mathcal{U}_h := \{\mathcal{Z}_{h,n} : u = 1, \ldots N_h\}$. To map every element $z \in \mathcal{Z}$ to its corresponding region we define the function function $\rho_h : \mathcal{Z} \to [N_h]$. The importance of these regions stays in the fact that there is a step-dependent feature map $\phi_h : \mathcal{Z} \to \mathbb{R}^{d_h}$ such that the optimal state-action value function is approximately linear in $\phi_h$ with a parameter depending on the region. Coherently, we define our approximator functions $\mathcal{Q}_h$ to be the set of functions on $\mathcal{Z}$ that are linear on the regions, namely

$$Q_h(z) = \phi_h(z)^\top \theta_{h, \rho_h(z)} \qquad z \in \mathcal{Z},$$

coherently, we define $\mathcal{V}_h := \{V(s) = \arg\max_a Q(s, a) : Q \in \mathcal{Q}_h\}$. By convenience, we will sometimes stack all these parameters in the list $\boldsymbol{\theta}_h = \{\theta_{h,n}\}_{n=1}^{N_h}$ and call $Q_h[\boldsymbol{\theta}_h]$ the corresponding state-action value function. In order not to allow the function set $\mathcal{Q}_h$ to contain functions with an arbitrary large slope, we impose the $\theta_{h,n}$ parameters to be constrained into bounded sets that we call $\mathcal{B}_{h,n}$.

We define the following bounds on the parameters of the process.

**Definition 4.** *We call*

- $L_\phi := \sup_{h \in [H], z \in \mathcal{Z}} \|\phi_h(z)\|_2$.

- $\mathcal{R}_{h,n} := \mathrm{diam}(\mathcal{B}_{h,n})$.

Now, calling $\mathcal{B}_h \subset \mathbb{R}^{d_h \times N_h}$ the sets given by

$$\mathcal{B}_h = \bigtimes_{n=1}^{N_h} \mathcal{B}_{h,n},$$

The performance of our algorithm will depend on the maximum approximation error that can be done by projecting the result of the Bellman operator on our functions $\mathcal{Q}$. In formula,

$$\sup_{\boldsymbol{\theta}_{h+1}\in\mathcal{B}_{h+1}} \inf_{\boldsymbol{\theta}_h\in\mathcal{B}_h} \|Q[\boldsymbol{\theta}_h](\cdot) - \mathcal{T}_h Q[\boldsymbol{\theta}_{h+1}](\cdot)\|_{L^\infty} \leq \mathcal{I}. \tag{9}$$

Note that the low inherent Bellman error MDPs are a subclass of this problem: it is sufficient to take $N_h = 1$ at every step.

## B.1 Definition of the quasi-optimal solution

In this subsection, we create what we will call a quasi-optimal solution for the MDP. Define $\boldsymbol{\theta}_h^\star$ in the following recursive way

$$\boldsymbol{\theta}_{h,n}^\star := \underset{\theta_n\in\mathcal{B}_{h,n}}{\arg\min} \max_{z\in\mathcal{Z}_{h,n}} |\phi_h(z)^\top \theta_n - \mathcal{T}_h Q_{h+1}[\boldsymbol{\theta}_{h+1}^\star](z)|, \tag{10}$$

So that $\boldsymbol{\theta}_h^\star$ is constructed by stacking these $\theta_{h,n}^\star$ terms. We have the following result.

**Theorem 6.** *The approximately optimal $Q-$function satisfies*

$$\forall h \qquad \|Q[\boldsymbol{\theta}_h^\star](\cdot) - Q_h^\star(\cdot)\|_{L^\infty} \leq (H - h)\mathcal{I}$$

*Proof.* We perform the proof by induction, with the case $h = H + 1$ being trivial.

$$\begin{aligned}
\|Q[\boldsymbol{\theta}_h^\star](\cdot) - Q_h^\star(\cdot)\|_{L^\infty} &= \|Q[\boldsymbol{\theta}_h^\star](\cdot) - \mathcal{T}_h Q_{h+1}[\boldsymbol{\theta}_{h+1}^\star](\cdot) + \mathcal{T}_h Q_{h+1}[\boldsymbol{\theta}_{h+1}^\star](\cdot) - Q_h^\star(\cdot)\|_{L^\infty} \\
&= \|Q[\boldsymbol{\theta}_h^\star](\cdot) - \mathcal{T}_h Q_{h+1}[\boldsymbol{\theta}_{h+1}^\star](\cdot) + \mathcal{T}_h Q_{h+1}[\boldsymbol{\theta}_{h+1}^\star](\cdot) - \mathcal{T}_h Q_{h+1}^\star(\cdot)\|_{L^\infty} \\
&\leq \|Q[\boldsymbol{\theta}_h^\star](\cdot) - \mathcal{T}_h Q_{h+1}[\boldsymbol{\theta}_{h+1}^\star](\cdot)\|_{L^\infty} \\
&\quad + \|\mathcal{T}_h Q_{h+1}[\boldsymbol{\theta}_{h+1}^\star](\cdot) - \mathcal{T}_h Q_{h+1}^\star(\cdot)\|_{L^\infty} \\
&\leq \|Q[\boldsymbol{\theta}_h^\star](\cdot) - \mathcal{T}_h Q_{h+1}[\boldsymbol{\theta}_{h+1}^\star](\cdot)\|_{L^\infty} + \|Q_{h+1}[\boldsymbol{\theta}_{h+1}^\star](\cdot) - Q_{h+1}^\star(\cdot)\|_{L^\infty},
\end{aligned}$$

where in the last passage we have used the non-expansivity of the Bellman operator. At this point, the second part is bounded as

$$\|Q_{h+1}[\boldsymbol{\theta}_{h+1}^\star](\cdot) - Q_{h+1}^\star(\cdot)\|_{L^\infty} \leq (H - h - 1)\mathcal{I},$$

by inductive hypothesis, while the first one satisfies, by equations (9) and (10)

$$\|Q[\boldsymbol{\theta}_h^\star](\cdot) - \mathcal{T}_h Q_{h+1}[\boldsymbol{\theta}_{h+1}^\star](\cdot)\|_{L^\infty} \leq \mathcal{I}.$$

Combining the two results gives the thesis. $\qquad\square$

## B.2 Algorithm

We call $s_h^k, a_h^k, r_h^k$ the state, action, and reward relative to step $h$ of episode $k$. Moreover, we denote $z_h^k := (s_h^k, a_h^k)$ and $\phi_h^k = \phi_h(z_h^k)$. We define, for any region $n$ step $h$ and episode $k$, the ridge regression matrix as

$$\Lambda_{h,n}^k := \sum_{\tau=1}^k \mathbf{1}\{\rho_h(z_h^\tau) = n\}\phi_h^\tau \phi_h^{\tau\top} + \lambda I.$$

Our algorithm, CINDERELLA (Algorithm 1), is built on top of an optimization problem (Eq. 7) which, at the start of each episode $k$, provides an optimistic version of the state-action value function. We recall it here:

$$\max_{\overline{\xi}_{h,n}, \overline{\theta}_{h,n}, \widehat{\theta}_{h,n}} \max_{a \in \mathcal{A}} \phi_1(s_1^k, a)^\top \overline{\theta}_{1, \rho(s_1^k, a)} \tag{11}$$

$$\text{s.t.} \quad \widehat{\theta}_{h,n} = \Lambda_{h,n}^{k}{}^{-1} \sum_{\tau=1}^{k-1} \mathbf{1}\{\rho_h(z_h^\tau) = n\} \phi_h^\tau (r_h^\tau + \overline{V}_{h+1}(s_{h+1}^\tau))$$

$$\overline{V}_{h+1}(\cdot) := \max_{a \in \mathcal{A}} \phi_{h+1}(\cdot, a)^\top \overline{\theta}_{h+1, \rho(\cdot, a)}$$

$$\overline{\theta}_{h,n} = \widehat{\theta}_{h,n} + \overline{\xi}_{h,n}$$

$$\|\overline{\xi}_{h,n}\|_{\Lambda_{h,n}^k} \leq \sqrt{\alpha_{h,n}^k}$$

Note that, with respect to the main paper, we have given a name to the quantity $\overline{V}_{h+1}(\cdot) := \max_{a \in \mathcal{A}} \phi_{h+1}(\cdot, a)^\top \overline{\theta}_{h+1, \rho(\cdot, a)}$ to make the variables more interpretable. The constant $\alpha_{h,n}^k$ will be defined in the next sections of this appendix.

## B.3 Algorithm analysis

We will now continue our work by proving that CINDERELLA is able to achieve a regret guarantee for this setting. First, we set up some additional notation.

**Additional notation** Collecting the solution of the optimization algorithm in the variables $\overline{\boldsymbol{\theta}}_h^k, \widehat{\boldsymbol{\theta}}_h^k$ and $\overline{\boldsymbol{\xi}}_h^k$, we define

$$\overline{Q}_h^k(z) := Q_h[\overline{\boldsymbol{\theta}}_h^k](z), \qquad \overline{V}_h^k(s) := \max_{a \in \mathcal{A}} \overline{Q}_h^k(s, a).$$

Moreover, in the rest of the section, once fixed a function $V : \mathcal{S} \to [0, 1]$, we define the Bellman error at step $h$ of episode $k$ as

$$\eta_h^k(V) := r_h^k - r_h(s_h^k, a_h^k) + V(s_{h+1}^k) - \mathbb{E}_{s' \sim p_h(\cdot | s_h^k, a_h^k)}[V(s')].$$

Furthermore, given a function $Q_{h+1} \in \mathcal{Q}_{h+1}$, we define

$$\mathring{\theta}_{h,n}(Q_{h+1}) := \underset{\theta_n \in \mathcal{B}_{h,n}}{\arg \min} \max_{z \in \mathcal{Z}_{h,n}} |\phi_h(z)^\top \theta_n - \mathcal{T}[Q_{h+1}](z)|, \tag{12}$$

and, as before, we collect all these vectors in $\mathring{\boldsymbol{\theta}}_h(Q_{h+1})$. Finally, we define

$$\Delta_h(Q_{h+1})(z) := \mathcal{T}_h Q_{h+1}(z) - Q[\mathring{\boldsymbol{\theta}}_h(Q_{h+1})](z).$$

## B.4 Decomposition of the estimated solution

We start by proving a proposition which establishes a relation between the variables of the optimization problem (11)

**Proposition 7.** *Let $\{\xi_{h,n}^k, \overline{\theta}_{h,n}^k, \widehat{\theta}_{h,n}^k\}_{h,n}$ be in the feasible region of Problem (11) at episode $k$ of the process. We have*

$$\overline{\theta}_{h,n}^k = \xi_{h,n}^k + \mathring{\theta}_{h,n}(\overline{Q}_{h+1}^k) + \underbrace{\Lambda_{h,n}^{k}{}^{-1}\sum_{\tau=1}^{k-1}\mathbf{1}\{\rho_h(z_h^\tau) = n\}\phi_h^\tau\Delta(\overline{Q}_{h+1}^k)(z_h^\tau)}_{T1_{h,n}^k} \tag{13}$$

$$- \underbrace{\Lambda_{h,n}^{k}{}^{-1}\mathring{\theta}_{h,n}(\overline{Q}_{h+1}^k)}_{T2_{h,n}^k} + \underbrace{\Lambda_{h,n}^{k}{}^{-1}\sum_{t=1}^{k-1}\mathbf{1}\{\rho_h(z_h^\tau) = n\}\phi_h^\tau\eta_h^\tau(\overline{V}_{h+1})}_{T3_{h,n}^k} \tag{14}$$

*Proof.* For simplicity, let us abbreviate $\rho_{h,n}^k := \mathbf{1}\{\rho_h(z_h^k) = n\}$. By construction,

$$\widehat{\theta}_{h,n}^k = \Lambda_{h,n}^{k}{}^{-1}\sum_{t=1}^{k-1}\rho_{h,n}^\tau\phi_h^\tau(r_h^\tau + \overline{V}_{h+1}^k(s_{h+1}^\tau))$$

$$= \Lambda_{h,n}^{k}{}^{-1}\sum_{\tau=1}^{k-1}\rho_{h,n}^\tau\phi_h^\tau\left(r_h(z_h^\tau) + \mathbb{E}_{s'\sim p_h(\cdot|z_h^\tau)}[\overline{V}_{h+1}^k(s')]\right) + \underbrace{\Lambda_{h,i}^{k}{}^{-1}\sum_{t=1}^{k-1}\rho_h^\tau\phi_h^\tau\eta_h^k(\overline{V}_{h+1}^k(s_{h+1}^\tau))}_{T3_{h,n}^k}$$

$$= \Lambda_{h,n}^{k}{}^{-1}\sum_{\tau=1}^{k-1}\rho_{h,n}^\tau\phi_h^\tau\mathcal{T}_h[\overline{Q}_{h+1}^k](z_h^\tau) + T3_{h,n}^k,$$

where we have used the definition of Bellman's optimality operator. By definition, we have

$$\mathcal{T}_h[\overline{Q}_{h+1}^k](z_h^\tau) = Q[\mathring{\boldsymbol{\theta}}_h(\overline{Q}_{h+1}^k)](z_h^\tau) + \Delta_h(\overline{Q}_{h+1}^k)(z_h^\tau),$$

which, when multiplied by $\rho_{h,n}^\tau$ lets only the term corresponding to the region $n$ remain. Then, by Equation (12), we have

$$\rho_{h,n}^\tau\mathcal{T}_h[\overline{Q}_{h+1}^k](z_h^\tau) = \rho_{h,n}^\tau\phi_h(z_h^\tau)^\top\mathring{\theta}_{h,n}(\overline{Q}_{h+1}^k) + \rho_{h,n}^\tau\Delta_h(\overline{Q}_{h+1}^k)(z_h^\tau).$$

Using this fact, we have

$$\Lambda_{h,n}^{k}{}^{-1}\sum_{\tau=1}^{k-1}\rho_{h,n}^\tau\phi_h^\tau\mathcal{T}_h[\overline{Q}_{h+1}^k](z_h^\tau) = \Lambda_{h,n}^{k}{}^{-1}\sum_{\tau=1}^{k-1}\rho_{h,n}^\tau\phi_h^\tau\left(\phi_h^{\tau\top}\mathring{\theta}_{h,n}(\overline{Q}_{h+1}^k) + \Delta_h(\overline{Q}_{h+1}^k)(z_h^\tau)\right)$$

$$= \Lambda_{h,n}^{k}{}^{-1}\sum_{\tau=1}^{k-1}\rho_{h,n}^\tau\phi_h^\tau\phi_h^{\tau\top}\mathring{\theta}_{h,n}(\overline{Q}_{h+1}^k) + T1_{h,n}^k.$$

For the remaining term, note that

$$\Lambda_{h,n}^{k}{}^{-1}\sum_{\tau=1}^{k-1}\rho_{h,n}^\tau\phi_h^\tau\phi_h^{\tau\top}\mathring{\theta}_{h,n}(\overline{Q}_{h+1}^k) = \Lambda_{h,n}^{k}{}^{-1}\left(-\lambda\mathring{\theta}_{h,n}(\overline{Q}_{h+1}^k) + \lambda\mathring{\theta}_{h,n}(\overline{Q}_{h+1}^k) + \sum_{\tau=1}^{k-1}\rho_{h,n}^\tau\phi_h^\tau\phi_h^{\tau\top}\mathring{\theta}_{h,n}(\overline{Q}_{h+1}^k)\right)$$

$$= \Lambda_{h,n}^{k}{}^{-1}\left(-\lambda\mathring{\theta}_{h,n}(\overline{Q}_{h+1}^k) + \Lambda_{h,n}^k\mathring{\theta}_{h,n}(\overline{Q}_{h+1}^k)\right)$$

$$= \mathring{\theta}_{h,n}(\overline{Q}_{h+1}^k) - T2_{h,n}^k$$

$\square$

The objective of the next lemmas is to show that the terms arising from the previous proposition are small with high probability.

**Lemma 1.** *For any $z \in \mathcal{Z}$, any time-step $h$ and any $n$, we have*

$$|\phi_h(z)^\top T1_{h,n}^k| \leq \|\phi_h(z)\|_{\Lambda_{h,n}^k}{}^{-1} \sqrt{p_{h,n}^k} \mathcal{I},$$

*where*

$$p_{h,n}^k := \sum_{\tau=1}^{k-1} \mathbf{1}\{\rho_h(z_h^\tau) = n\},$$

*and in particular*

$$\|T1_{h,n}^k\|_{\Lambda_{h,n}^k} \leq \sqrt{p_{h,n}^k} \mathcal{I}.$$

*Proof.* By definition,

$$|\phi_h(z)^\top T1_{h,n}^k| = \left| \phi_h(z)^\top \Lambda_{h,n}^k{}^{-1} \sum_{\tau=1}^{k-1} \mathbf{1}\{\rho_h(z_h^\tau) = n\} \phi_h^\tau \Delta(\overline{Q}_{h+1}^k)(z_h^\tau) \right|$$

$$\leq \|\Lambda_{h,n}^k{}^{-1} \phi_h(s,a)\|_{\Lambda_{h,n}^k} \left\| \sum_{\tau=1}^{k-1} \mathbf{1}\{\rho_h(z_h^\tau) = n\} \phi_h^\tau \Delta(\overline{Q}_{h+1}^k)(z_h^\tau) \right\|_{\Lambda_{h,n}^k{}^{-1}}$$

$$= \|\phi_h(s,a)\|_{\Lambda_{h,n}^k{}^{-1}} \left\| \sum_{\tau=1}^{k-1} \mathbf{1}\{\rho_h(z_h^\tau) = n\} \phi_h^\tau \Delta(\overline{Q}_{h+1}^k)(z_h^\tau) \right\|_{\Lambda_{h,n}^k{}^{-1}}.$$

At this point, leave the first term and, in the second we rewrite the sum for the indices where the indicator function is not zero, getting

$$\left\| \sum_{\tau'=1}^{p_{h,n}^k} \phi_h^{\tau'} \Delta(\overline{Q}_{h+1}^k)(z_h^{\tau'}) \right\|_{\Lambda_{h,n}^k{}^{-1}}.$$

We can then use Lemma 8 from [42], taking $a_i = \phi_h^{\tau'}$ and $b_i = \Delta(\overline{Q}_{h+1}^k)(z_h^{\tau'})$. As $|b_i| \leq \mathcal{I}$, by the low inherent Bellman error assumption (Eq. 9), this gives

$$\left\| \sum_{\tau'=1}^{p_{h,n}^k} \phi_h^{\tau'} \Delta(\overline{Q}_{h+1}^k)(z_h^{\tau'}) \right\|_{\Lambda_{h,n}^k{}^{-1}} \leq \sqrt{p_{h,n}^k} \mathcal{I},$$

which ends the proof of the first part. The second one comes from

$$\|T1_{h,n}^k\|_{\Lambda_{h,n}^k} = \left\| \Lambda_{h,n}^k{}^{-1} \sum_{\tau'=1}^{p_{h,n}^k} \phi_h^{\tau'} \Delta(\overline{Q}_{h+1}^k)(z_h^{\tau'}) \right\|_{\Lambda_{h,n}^k}$$

$$= \left\| \sum_{\tau'=1}^{p_{h,n}^k} \phi_h^{\tau'} \Delta(\overline{Q}_{h+1}^k)(z_h^{\tau'}) \right\|_{\Lambda_{h,n}^k{}^{-1}}$$

$$\leq \sqrt{p_{h,n}^k} \mathcal{I}.$$

$\square$

We proceed bounding the second part of the sum.

**Lemma 2.** *For any $z \in \mathcal{Z}$, any time-step $h$ and any $n$, we have*

$$|\phi_h(z)^\top T2_{h,n}^k| \leq \|\phi_h(z)\|_{\Lambda_{h,n}^k}{}^{-1} \lambda^{-1} \mathcal{R}_{h,n},$$

*and, in particular*

$$\|T2_{h,n}^k\|_{\Lambda_{h,n}^k} \leq \lambda^{-1} \mathcal{R}_{h,n}.$$

*Proof.* By definition,

$$|\phi_h(z)^\top T2^k_{h,n}| = \left|\phi_h(z)^\top \Lambda^k_{h,n}{}^{-1}\overset{\circ}{\theta}_{h,n}(\overline{Q}^k_{h+1})\right|$$

$$\leq \|\Lambda^k_{h,n}{}^{-1}\phi_h(z)\|_{\Lambda^k_{h,n}}\|\overset{\circ}{\theta}_{h,n}(\overline{Q}^k_{h+1})\|_{\Lambda^k_{h,n}{}^{-1}}$$

$$= \|\phi_h(z)\|_{\Lambda^k_{h,n}{}^{-1}}\|\overset{\circ}{\theta}_{h,n}(\overline{Q}^k_{h+1})\|_{\Lambda^k_{h,n}{}^{-1}}$$

$$\leq \|\phi_h(z)\|_{\Lambda^k_{h,n}{}^{-1}}\lambda^{-1}\|\overset{\circ}{\theta}_{h,n}(\overline{Q}^k_{h+1})\|_2$$

$$\leq \|\phi_h(z)\|_{\Lambda^k_{h,n}{}^{-1}}\lambda^{-1}\mathcal{R}_{h,n},$$

where the second inequality comes from the fact that $\Lambda^k_{h,n}$ is the sum of $\lambda I$ and a positive semi-definite matrix, and the last one from the fact that $\overset{\circ}{\theta}_{h,n}(\overline{Q}^k_{h+1}) \in \mathcal{B}_{h,n}$. The second part comes from

$$\|T2^k_{h,n}\|_{\Lambda^k_{h,n}} = \|\Lambda^k_{h,n}{}^{-1}\overset{\circ}{\theta}_{h,n}(\overline{Q}^k_{h+1})\|_{\Lambda^k_{h,n}} = \|\overset{\circ}{\theta}_{h,n}(\overline{Q}^k_{h+1})\|_{\Lambda^k_{h,n}{}^{-1}} \leq \lambda^{-1}\mathcal{R}_{h,n}.$$

$\square$

The last part of the sum is more complex and, in order to bound it, we need to define a failure event.

## B.5 Failure event

For every $h, k, n$ we define the failure event in the following way:

$$F^k_{h,n} := \left\{\exists V \in \mathcal{V}_h : \left\|\sum_{t=1}^{k-1}\mathbf{1}\{\rho_h(z^\tau_h) = n\}\phi^\tau_h\eta^\tau_h(V)\right\|_{\Lambda^k_{h,n}{}^{-1}} > \sqrt{\beta^k_{h,n}}\right\},$$

for a threshold $\sqrt{\beta^k_{h,n}}$ to be defined. The first step to bound the probability of this event is to compute the covering number of the function space $\mathcal{V}_h$

**Proposition 8.** *The $\varepsilon-$covering number of $\mathcal{V}_h$ in infinity norm satisfies*
$$\log\mathcal{N}(\varepsilon, \mathcal{V}_h) \leq \mathcal{O}\left(Nd\log\left(\mathcal{R}_{h,\max}/\varepsilon\right)\right),$$

*where $\mathcal{R}_{h,\max} := \sup_{n\in[N_h]}\mathcal{R}_{h,n}$.*

*Proof.* Note that the covering number of $\mathcal{V}_h$ is not higher than the one of $\mathcal{Q}_h$. Indeed, let $\mathcal{Q}^\varepsilon_h$ be a $\varepsilon-$cover for $\mathcal{Q}_h$ and define
$$\mathcal{V}^\varepsilon_h := \{V(s) = \max_{a\in\mathcal{A}}Q(s,a) : Q \in \mathcal{Q}^\varepsilon_h\}.$$

Then, taking any $V \in \mathcal{V}_h$ there is $Q \in \mathcal{Q}_h$ such that $V(s) = \max_{a\in\mathcal{A}}Q(s,a)$. At this point, taking
$$\widehat{V}(s) = \max_{a\in\mathcal{A}}\widehat{Q}(s,a),$$

where $\widehat{Q} \in \mathcal{Q}^\varepsilon_h$ such that $\|Q - \widehat{Q}\|_\infty \leq \varepsilon$, we have that $\widehat{V} \in \mathcal{V}^\varepsilon_h$ by definition and
$$\|Q - \widehat{Q}\|_\infty \leq \|V - \widehat{V}\|_\infty \leq \varepsilon.$$

Now, the question is reduced to covering $\mathcal{Q}_h$. by definition, every $Q_h \in \mathcal{Q}_h$ takes the form
$$Q_h(z) = \phi_h(z)^\top\theta_{h,\rho_h(z)},$$

where $\rho_h : \mathcal{Z} \to [N_h]$ and every vector $\theta_{h,n}$ is $d_h-$dimensional with the norm bounded by $\mathcal{R}_{h,n}$. As the domain $\mathcal{Z}$ is partitioned into regions $\mathcal{Z}_{h,n}$, if we get a family of coverings $\mathcal{Q}^\varepsilon_{h,n}$ that cover each function in $\mathcal{Q}_h$ restricted to the set $\mathcal{Z}_{h,n}$, we can obtain a covering of $\mathcal{Q}_h$ by defining it in this way

$$\mathcal{Q}_h^\varepsilon := \left\{ Q : Q|_{\mathscr{Z}_{h,n}} = Q_n \qquad Q_n \in \mathcal{Q}_{h,n}^\varepsilon \right\},$$

i.e., we cover every function $Q$ by taking the nearest cover function on each region. As a result, the total number of elements in $\mathcal{Q}_h^\varepsilon$ corresponds to the product $\prod_{n=1}^{N_h} |\mathcal{Q}_{h,n}^\varepsilon|$ of the elements in the smaller sets, so that we have only to estimate $|\mathcal{Q}_{h,n}^\varepsilon|$ for every $n$. To this aim, note that the functions in this set are all linear, as we are restricting to $\mathscr{Z}_{h,n}$, so that, applying a standard bound for the covering number of spaces of linear functions we can find $\mathcal{Q}_{h,n}^\varepsilon$ such that

$$|\mathcal{Q}_{h,n}^\varepsilon| = \mathcal{O}\left( (\mathcal{R}_{h,n}/\varepsilon)^{d_h} \right).$$

Therefore, the total covering size amounts to

$$|\mathcal{Q}_h^\varepsilon| = \mathcal{O}\left( ( \sup_{n \in [N_h]} \mathcal{R}_{h,n}/\varepsilon)^{Nd} \right),$$

which completes the proof. $\qquad\square$

**Theorem 9.** *For a choice of*
$$\sqrt{\beta_{h,n}^k} = \tilde{\mathcal{O}}\left( \sqrt{d_h + \log(\mathcal{N}(1/\sqrt{k}, \mathcal{V}_h)) + \log(1/\delta)} \right),$$
*we have* $\mathbb{P}(F_{h,n}^k) \le \delta$.

*Proof.* Let $\mathcal{V}_h^\varepsilon$ be a $\varepsilon$ cover of $\mathcal{V}_h$ in infinity norm, so that $|\mathcal{V}_h^\varepsilon| = \mathcal{N}(\varepsilon, \mathcal{V}_h)$. Now, for every $V \in \mathcal{V}_h$ we call
$$\widetilde{V} \in \mathcal{V}_h^\varepsilon : \; \|\widetilde{V} - V\|_{L^\infty} \le \varepsilon.$$

In this way, we have

$$\left\| \sum_{t=1}^{k-1} \rho_{h,n}^\tau \phi_h^\tau \eta_h^\tau(V) \right\|_{\Lambda_{h,n}^k}^{-1} = \left\| \sum_{t=1}^{k-1} \rho_{h,n}^\tau \phi_h^\tau \left( r_h^k - r_h(s_h^k, a_h^k) + V(s_{h+1}^k) - \mathbb{E}_{s' \sim p_h(\cdot|s_h^k, a_h^k)}[V(s')] \right) \right\|_{\Lambda_{h,n}^k}^{-1}$$

$$\le \left\| \sum_{t=1}^{k-1} \rho_{h,n}^\tau \phi_h^\tau \left( r_h^k - r_h(s_h^k, a_h^k) + \widetilde{V}(s_{h+1}^k) - \mathbb{E}_{s' \sim p_h(\cdot|s_h^k, a_h^k)}[\widetilde{V}(s')] \right) \right\|_{\Lambda_{h,n}^k}^{-1}$$

$$+ \left\| \sum_{t=1}^{k-1} \rho_{h,n}^\tau \phi_h^\tau \left( V(s_{h+1}^k) - \widetilde{V}(s_{h+1}^k) \right) \right\|_{\Lambda_{h,n}^k}^{-1}$$

$$+ \left\| \sum_{t=1}^{k-1} \rho_{h,n}^\tau \phi_h^\tau \left( \mathbb{E}_{s' \sim p_h(\cdot|s_h^k, a_h^k)}[\widetilde{V}(s') - V(s')] \right) \right\|_{\Lambda_{h,n}^k}^{-1}.$$

As we have
$$|V(s_{h+1}^k) - \widetilde{V}(s_{h+1}^k)| \le \varepsilon, \qquad |\mathbb{E}_{s' \sim p_h(\cdot|s_h^k, a_h^k)}[\widetilde{V}(s') - V(s')]| \le \varepsilon,$$

the same argument used in the proof Lemma 1, allows us to bound the last two terms with $2\sqrt{p_{h,n}^k}\varepsilon$.

The remaining term corresponds, for fixed $\widetilde{V}$ and indicating with $\mathcal{F}_h^k$ the filtration generated by all the events of the process up to step $h$ of episode $k$, to

$$\left\| \sum_{t=1}^{k-1} \underbrace{\rho_{h,n}^\tau \phi_h^\tau}_{\mathcal{F}_h^k-meas.} \underbrace{\left( r_h^k - r_h(s_h^k, a_h^k) + \widetilde{V}(s_{h+1}^k) - \mathbb{E}_{s' \sim p_h(\cdot|s_h^k, a_h^k)}[\widetilde{V}(s')] \right)}_{\mathbb{E}[\cdot|\mathcal{F}_h^k]=0} \right\|_{\Lambda_{h,n}^k}^{-1},$$

where the first term is an $\mathbb{R}^{d_h}-$valued stochastic process which is entirely determined by the current state and current action, while the second is zero-mean conditioned on the rest of the process, and $1-$subgaussian thanks to assumption 1. This last fact allows to apply Theorem 1 from [1], obtaining with probability at least $1 - \delta$, for fixed $\widetilde{V}$,

$$\left\| \sum_{t=1}^{k-1} \rho_{h,n}^\tau \phi_h^\tau \left( r_h^k - r_h(s_h^k, a_h^k) + \widetilde{V}(s_{h+1}^k) - \mathbb{E}_{s' \sim p_h(\cdot | s_h^k, a_h^k)}[\widetilde{V}(s')] \right) \right\|_{\Lambda_{h,n}^k{}^{-1}}^2$$

$$\leq 2 \log \left( \frac{\det \lambda I^{-1/2} \det \Lambda_{h,n}^k{}^{1/2}}{\delta} \right) \leq d_h \log(\lambda^{-1}) + d_h \log((1 + kL_\phi)) + \log(\delta^{-1}).$$

Applying a union bound, it follows that the same result holds for every function in $\mathcal{V}_h^\varepsilon$ if we replace $\log(\delta^{-1})$ with $\log(\mathcal{N}(\varepsilon, \mathcal{V}_h)/\delta)$. Merging this result with what we have found before, we have that, with probability at least $1 - \delta$, for all $V \in \mathcal{V}_h$, we have

$$\left\| \sum_{t=1}^{k-1} \rho_{h,n}^\tau \phi_h^\tau \eta_h^\tau(V) \right\|_{\Lambda_{h,n}^k{}^{-1}} \leq \sqrt{d_h \log(\lambda^{-1}) + d_h \log((1 + kL_\phi)) + \log(\mathcal{N}(\varepsilon, \mathcal{V}_h)) + \log(1/\delta)}$$

$$+ 2\sqrt{p_{h,n}^k \varepsilon}.$$

If we just bound $\sqrt{p_{h,n}^k} \leq \sqrt{k}$, we can take $\varepsilon = 1/\sqrt{k}$ to have that the previous is bounded by

$$\sqrt{\beta_{h,n}^k} := \sqrt{d_h \log(\lambda^{-1}) + d_h \log((1 + kL_\phi)) + \log(\mathcal{N}(1/\sqrt{k}, \mathcal{V}_h)) + \log(1/\delta) + 2},$$

which completes the proof. $\qquad\square$

From this theorem, a simple corollary follows by just taking a union bound over $h \in [H], n \in [N_h], k \in [K]$:

**Corollary 10.** *For a choice of*

$$\sqrt{\beta_{h,n}^k} = \widetilde{\mathcal{O}} \left( \sqrt{d_h + \log(\mathcal{N}(1/\sqrt{k}, \mathcal{V}_h)) + \log(N_h H K/\delta))} \right),$$

*we have*

$$\mathbb{P} \left( \bigcup_{h,n,k} F_{h,n}^k \right) \leq \delta.$$

From now on, we are going to indicate the good event $E$ as the opposite of the event defined in the previous corollary

$$E := \bigcap_{h,n,k} (F_{h,n}^k)^c,$$

moreover, $\sqrt{\beta_{h,n}^k}$ will always be the quantity defined by Corollary 10. We conclude the section proving the following

**Lemma 3.** *Under the event $E$, for any $z \in \mathcal{Z}$, any time-step $h$ of episode $k$ and any $n$, we have*

$$|\phi_h(z)^\top T3_{h,n}^k| \leq \|\phi_h(z)\|_{\Lambda_{h,n}^k{}^{-1}} \sqrt{\beta_{h,n}^k},$$

*and, in particular*

$$\|T3_{h,n}^k\|_{\Lambda_{h,n}^k{}^{-1}} \leq \sqrt{\beta_{h,n}^k}.$$

*Proof.* By definition,

$$
\begin{aligned}
|\phi_h(z)^\top T3_{h,n}^k| &= \left| \phi_h(z)^\top \Lambda_{h,n}^{k}{}^{-1} \sum_{t=1}^{k-1} \mathbf{1}\{\rho_h(z_h^\tau) = n\}\phi_h^\tau \eta_h^\tau (\overline{V}_{h+1}) \right| \\
&\leq \left\| \Lambda_{h,n}^{k}{}^{-1}\phi_h(z) \right\|_{\Lambda_{h,n}^k} \left\| \sum_{t=1}^{k-1} \mathbf{1}\{\rho_h(z_h^\tau) = n\}\phi_h^\tau \eta_h^\tau (\overline{V}_{h+1}) \right\|_{\Lambda_{h,n}^{k}{}^{-1}} \\
&= \left\| \phi_h(z) \right\|_{\Lambda_{h,n}^{k}{}^{-1}} \left\| \sum_{t=1}^{k-1} \mathbf{1}\{\rho_h(z_h^\tau) = n\}\phi_h^\tau \eta_h^\tau (\overline{V}_{h+1}) \right\|_{\Lambda_{h,n}^{k}{}^{-1}}.
\end{aligned}
$$

Under the good event $E$, we have that the second term is bounded by $\sqrt{\beta_{h,n}^k}$, for the choice defined in Corollary 10. The second part comes from the fact that

$$
\begin{aligned}
\|T3_{h,n}^k\|_{\Lambda_{h,n}^k} &= \left\| \Lambda_{h,n}^{k}{}^{-1} \sum_{t=1}^{k-1} \mathbf{1}\{\rho_h(z_h^\tau) = n\}\phi_h^\tau \eta_h^\tau (\overline{V}_{h+1}) \right\|_{\Lambda_{h,n}^k} \\
&= \left\| \sum_{t=1}^{k-1} \mathbf{1}\{\rho_h(z_h^\tau) = n\}\phi_h^\tau \eta_h^\tau (\overline{V}_{h+1}) \right\|_{\Lambda_{h,n}^{k}{}^{-1}} \\
&\leq \sqrt{\beta_{h,n}^k}.
\end{aligned}
$$

$\square$

### B.6 Quasi-optimal solution and good event

Thanks to the previous result, we are able to show that the quasi-optimal solution is feasible with high probability.

**Theorem 11.** *Consider the optimization problem* (11), *where* $\sqrt{\alpha_{h,n}^k}$ *is set to* $\sqrt{\beta_{h,n}^k} + \sqrt{p_{h,n}^k} + \lambda^{-1}\mathcal{R}_{h,n}$. *Then, under the good event* $E$, *the quasi-optimal solution* $\{\boldsymbol{\theta}_h^\star\}_{h=1}^H$ *(see* (10)*) is feasible for* (11) *at any episode* $k$.

*Proof.* We perform the proof by induction on the time-step $h$, starting from $h = H$. Assuming that for every $n \in [N_h]$ the choice $\theta_{h+1,n}^\star$ is feasible we have that the choice

$$
\overline{Q}_{h+1}(z) := Q_{h+1}[\boldsymbol{\theta}_{h+1}^\star](z)
$$

is also feasible. With this choice, proposition 7 ensures that, for every $n$,

$$
\overline{\theta}_{h,n} = \overline{\xi}_{h,n} + \overset{\circ}{\theta}_{h,n}(Q_{h+1}[\boldsymbol{\theta}_{h+1}^\star]) + T1_{h,n}^k - T2_{h,n}^k + T3_{h,n}^k. \tag{15}
$$

Now, to show that also $\boldsymbol{\theta}_h^\star$ is feasible, we have to prove that, substituting each $\overline{\theta}_{h,n} = \theta_{h,n}^\star$ in the previous equation, we get a value for $\overline{\xi}_{h,n}$ such that $\|\overline{\xi}_{h,n}\|_{\Lambda_{h,n}^k} \leq \sqrt{\alpha_{h,n}^k}$.

First note that, by definition, $\theta_{h,n}^\star = \overset{\circ}{\theta}_{h,n}(Q_{h+1}[\boldsymbol{\theta}_{h+1}^\star])$, so that the previous equation simplifies to

$$
-\overline{\xi}_{h,n} = T1_{h,n}^k - T2_{h,n}^k + T3_{h,n}^k. \tag{16}
$$

At this point we have by triangular inequality

$$
\|\overline{\xi}_{h,n}\|_{\Lambda_{h,n}^k} \leq \|T1_{h,n}^k\|_{\Lambda_{h,n}^k} + \|T2_{h,n}^k\|_{\Lambda_{h,n}^k} + \|T3_{h,n}^k\|_{\Lambda_{h,n}^k} \tag{17}
$$

$$
\leq \sqrt{p_{h,n}^k}\mathcal{I} + \lambda^{-1}\mathcal{R}_{h,n} + \sqrt{\beta_{h,n}^k} \tag{18}
$$

$$
= \sqrt{\alpha_{h,n}^k}. \tag{19}
$$

where the second comes from Lemmas 1, 2 and 3. This completes the proves that each $\theta_{h,n}^\star$ is feasible, meaning that also $\boldsymbol{\theta}_h^\star$ is, and completes the inductive step. $\qquad\square$

From the feasibility of $\boldsymbol{\theta}_h^\star$ a simple corollary follows

**Corollary 12.** *Under the good event $E$, at each episode $k$, the $V-$function $\overline{V}_1^k(\cdot)$ estimated from $\overline{\boldsymbol{\theta}}_1$ as solution of solution of the optimization problem* (11) *satisfies*

$$\forall s \in \mathcal{S} \qquad V_1^\star(s) - \overline{V}_1^k(s) \leq (H-1)\mathcal{I}.$$

*Proof.* By design of the optimization problem, and the fact $\{\boldsymbol{\theta}_h^\star\}_{h=1}^H$ is feasible under $E$, we have

$$\overline{V}_1^k(s) \geq V[\boldsymbol{\theta}_1^\star](s).$$

Now, we have

$$
\begin{aligned}
V_1^\star(s) - \overline{V}_1^k(s) &\leq V_1^\star(s) - V[\boldsymbol{\theta}_1^\star](s) \\
&= \max_{a \in \mathcal{A}} Q_1^\star(s,a) - \max_{a \in \mathcal{A}} Q[\boldsymbol{\theta}_1^\star](s,a) \\
&\leq \max_{a \in \mathcal{A}} Q_1^\star(s,a) - Q[\boldsymbol{\theta}_1^\star](s,a) \\
&\leq \|Q_1^\star(\cdot) - Q[\boldsymbol{\theta}_1^\star](\cdot)\|_{L^\infty} \leq (H-1)\mathcal{I},
\end{aligned}
$$

where the last passage is possible by theorem 6. $\qquad\square$

Moreover, we are also able to prove that the solution found by the optimization algorithm (11) is an "almost fixed" point of the Bellman optimiality operator.

**Proposition 13.** *Under the good event $E$, the $Q-$function computed with the optimization algorithm at each step $k$ satisfies Outside of the failure event, we have*

$$\forall z \in \mathcal{Z} \qquad \left|\overline{Q}_h^k(z) - \mathcal{T}_h[\overline{Q}_{h+1}^k](z)\right| \leq \mathcal{I} + 2\sqrt{\alpha_{h,\rho_h(z)}^k}\|\phi_h(z)\|_{\Lambda_{h,\rho_h(z)}^k{}^{-1}}.$$

*Proof.* By Proposition 7, we have, taking $n := \rho_h(z)$,

$$
\begin{aligned}
|\overline{Q}_h^k(z) - \mathcal{T}_h[\overline{Q}_{h+1}^k](z)| &\leq |\phi_h(z)^\top \overline{\theta}_{h,n} - \mathcal{T}_h[\overline{Q}_{h+1}](z)| \\
&\quad + \left|\phi_h(z)^\top(\overline{\xi}_{h,n} + T1_{h,n}^k - T2_{h,n}^k + T3_{h,n}^k)\right| \\
&\leq \mathcal{I} + \left|\phi_h(z)^\top \overline{\xi}_{h,n}\right| + \left|\phi_h(z)^\top T1_{h,n}^k\right| \\
&\quad + \left|\phi_h(z)^\top T2_{h,n}^k\right| + \left|\phi_h(z)^\top T3_{h,n}^k\right|.
\end{aligned}
$$

The first term satisfies, by the constrain in the program,

$$\left|\phi_h(s,a)^\top \overline{\xi}_{h,n}\right| \leq \|\overline{\xi}_{h,n}\|_{\Lambda_{h,n}^k}\|\phi_h(s,a)\|_{\Lambda_{h,n}^k{}^{-1}} \leq \sqrt{\alpha_{h,n}^k}\|\phi_h(s,a)\|_{\Lambda_{h,n}^k{}^{-1}}.$$

The other terms satisfy, thanks to lemmas 1,2,3, satisfies

$$\left|\phi_h(z)^\top T1_{h,n}^k\right| + \left|\phi_h(z)^\top T2_{h,n}^k\right| + \left|\phi_h(z)^\top T3_{h,n}^k\right| \leq \sqrt{\alpha_{h,n}^k}\|\phi_h(s,a)\|_{\Lambda_{h,n}^k{}^{-1}}.$$

$\qquad\square$

## C  Regret bound

Using all the results that we have proved, we can finally bound the regret

**Theorem 14.** *Under event $E$,* CINDERELLA, *for $\lambda = 1$ achieves a regret bound of order*

$$R_K \leq \widetilde{\mathcal{O}}\left(\sum_{h=1}^H \sqrt{KN_h}\left((L_\phi + \sqrt{d_h})\sup_n \mathcal{R}_{h,n} + d_h + \sqrt{d_h \log(\mathcal{N}(1/\sqrt{K}, \mathcal{V}_h))}\right) + KH\mathcal{I}\sqrt{d_h}\right).$$

*Proof.* By Corollary 12, the regret is bounded by

$$R_K = \sum_{k=1}^{K} V_1^*(s_1^k) - V_1^{\pi_k}(s_1^k) \le KH\mathcal{I} + \sum_{k=1}^{K} \overline{V}_1^k(s_1^k) - V_1^{\pi_k}(s_1^k).$$

Note that, under the good event,

$$\overline{V}_h^k(s_h^k) - V_h^{\pi_k}(s_h^k) = \overline{Q}_h^k(z_h^k) - V_h^{\pi_k}(s_h^k)$$
$$\overset{\text{Prop. 13}}{\le} \mathcal{I} + 2\sqrt{\alpha_{h,\rho_h(z_h^k)}^k}\|\phi_h(z_h^k)\|_{\Lambda_{h,\rho_h(z_h^k)}^k}^{-1} + \mathcal{T}_h[\overline{Q}_{h+1}^k](z_h^k) - V_h^{\pi_k}(s_h^k)$$
$$= \mathcal{I} + 2\sqrt{\alpha_{h,\rho_h(z_h^k)}^k}\|\phi_h(z_h^k)\|_{\Lambda_{h,\rho_h(z_h^k)}^k}^{-1} + \mathbb{E}_{s'\sim p_h(\cdot|z_h^k)}[\overline{V}_{h+1}^k(s') - V_{h+1}^{\pi_k}(s')].$$

Now, if we define the quantity

$$\zeta_h^k := \mathbb{E}_{s'\sim p_h(\cdot|s_h^k,a_h^k)}[\overline{V}_{h+1}^k(s') - V_{h+1}^{\pi_k}(s')] - \overline{V}_{h+1}^k(s_{h+1}^k) + V_{h+1}^{\pi_k}(s_{h+1}^k),$$

we can rewrite the previous relation as a telescopic sum:

$$\overline{V}_1^k(s_1^k) - V_1^{\pi_k}(s_1^k) \le \mathcal{I} + 2\sqrt{\alpha_{1,\rho_h(z_1^k)}^k}\|\phi_1(z_1^k)\|_{\Lambda_{1,\rho_1(z_1^k)}^k}^{-1} + \mathbb{E}_{s'\sim p_1(\cdot|z_1^k)}[\overline{V}_2^k(s') - V_2^{\pi_k}(s')]$$
$$= \mathcal{I} + 2\sqrt{\alpha_{1,\rho_h(z_1^k)}^k}\|\phi_1(z_1^k)\|_{\Lambda_{1,\rho_1(z_1^k)}^k}^{-1} + \zeta_1^k + \underbrace{\overline{V}_2^k(s_2^k) - V_2^{\pi_k}(s_2^k)}_{\text{same quantity at next time step}}$$
$$\le H\mathcal{I} + \sum_{h=1}^{H} 2\sqrt{\alpha_{h,\rho_h(z_h^k)}^k}\|\phi_h(z_h^k)\|_{\Lambda_{h,\rho_h(z_h^k)}^k}^{-1} + \zeta_h^k.$$

This equation allows us to bound the regret, under the good event, as

$$R_K \le KH\mathcal{I} + \sum_{k=1}^{K}\sum_{h=1}^{H} 2\sqrt{\alpha_{h,\rho_h(z_h^k)}^k}\|\phi_h(z_h^k)\|_{\Lambda_{h,\rho_h(z_h^k)}^k}^{-1} + \zeta_h^k.$$

We start from the last term, which we rewrite, by exchanging the index of the sums as

$$\sum_{k=1}^{K}\sum_{h=1}^{H} \zeta_h^k = \sum_{h=1}^{H}\sum_{k=1}^{K} \zeta_h^k.$$

For fixed $h$ the term $\zeta_h^k$ is a martingale difference and, if it is bounded almost surely in $[-C,C]$ for some constant $C > 0$, we can apply Hoeffding's inequality to prove that w.p. at least $1 - \delta$,

$$-2C\sqrt{K\log(1/\delta)} \le \sum_{k=1}^{K} \zeta_h^k \le 2C\sqrt{K\log(1/\delta)}.$$

In our case, due to our problem definition we have that $C$ may be chosen as $L_\phi \sup_n \mathcal{R}_{h,n}$. Imposing that the previous inequality works at every step at the same time, and summing over $h$, we get, w.p. $1 - \delta$,

$$\sum_{h=1}^{H}\sum_{k=1}^{K} \zeta_h^k \le 2\sqrt{K\log(H/\delta)} \sum_{h=1}^{H} L_\phi \sup_n \mathcal{R}_{h,n}. \tag{20}$$

Now, we go on bounding the other term

$$\sum_{k=1}^{K}\sum_{h=1}^{H} 2\sqrt{\alpha_{h,\rho_h(z_h^k)}^k}\|\phi_h(z_h^k)\|_{\Lambda_{h,\rho_h(z_h^k)}^k{}^{-1}} = \sum_{h=1}^{H}\sum_{k=1}^{K} 2\sqrt{\alpha_{h,\rho_h(z_h^k)}^k}\|\phi_h(z_h^k)\|_{\Lambda_{h,\rho_h(z_h^k)}^k{}^{-1}}$$

$$\leq \sum_{h=1}^{H}\sqrt{K}\sqrt{\sum_{k=1}^{K} 4\alpha_{h,\rho_h(z_h^k)}^k\|\phi_h(z_h^k)\|_{\Lambda_{h,\rho_h(z_h^k)}^k{}^{-1}}^2} \quad (21)$$

Where the last passage is due to the Cauchy-Schwartz inequality. The last sum can be rewritten by summing over the regions instead of the episodes:

$$\sum_{k=1}^{K} 4\alpha_{h,\rho_h(z_h^k)}^k\|\phi_h(z_h^k)\|_{\Lambda_{h,\rho_h(z_h^k)}^k{}^{-1}}^2 = \sum_{n=1}^{N_h}\sum_{i_n=1}^{p_{h,n}^K} 4\alpha_{h,n}^{i_n}\|\phi_h(z_h^{i_n})\|_{\Lambda_{h,n}^{i_n}{}^{-1}}^2,$$

where the index $i_n$ enumerates the episodes $k$ where $z_h^k \in \mathcal{Z}_{h,n}$. At this point, recall that we have set $\sqrt{\alpha_{h,n}^k} = \sqrt{\beta_{h,n}^k} + \sqrt{p_{h,n}^k} + \lambda^{-1}\mathcal{R}_{h,n}$, so that, in particular

$$\alpha_{h,n}^k \leq 3\beta_{h,n}^k + 3p_{h,n}^k + 3\lambda^{-1}\mathcal{R}_{h,n}$$
$$\leq 3p_{h,n}^k\mathcal{I}^2 + 3d + 3\log(\mathcal{N}(1/\sqrt{k}, \mathcal{V}_h)) + 3\log(1/\delta) + 3\lambda^{-1}\mathcal{R}_{h,n}.$$

This way, the previous sum $\sum_{n=1}^{N_h}\sum_{i_n=1}^{p_{h,n}^K} 4\alpha_{h,n}^{i_n}\|\phi_h(z_h^{i_n})\|_{\Lambda_{h,n}^{i_n}{}^{-1}}^2$ can be again decomposed in few terms, according to the decomposition of $\alpha_{h,n}^k$.

$$12\sum_{n=1}^{N_h}\sum_{i_n=1}^{p_{h,n}^K} p_{h,n}^K\mathcal{I}^2\|\phi_h(z_h^{i_n})\|_{\Lambda_{h,n}^{i_n}{}^{-1}}^2 = 12\sum_{n=1}^{N_h} p_{h,n}^K\mathcal{I}^2\sum_{i_n=1}^{p_{h,n}^K}\|\phi_h(z_h^{i_n})\|_{\Lambda_{h,n}^{i_n}{}^{-1}}^2.$$

Note that, by construction of the matrix $\Lambda_{h,n}^{i_n}$, we can use Lemma 11 from [1] to have

$$\sum_{i_n=1}^{p_{h,n}^K}\|\phi_h(z_h^{i_n})\|_{\Lambda_{h,n}^{i_n}{}^{-1}}^2 \leq 2d_h\log((\lambda + p_{h,n}^K L_\phi^2/d_h)) - \log(\det(\lambda I))),$$

where by fixing $\lambda = 1$ we have

$$\sum_{i_n=1}^{p_{h,n}^K}\|\phi_h(z_h^{i_n})\|_{\Lambda_{h,n}^{i_n}{}^{-1}}^2 \leq 2d_h\log((1 + p_{h,n}^K L_\phi^2/d_h)) \leq 2d_h\log(KL_\phi^2 + 1).$$

Therefore, we can bound the whole term as

$$12\sum_{n=1}^{N_h}\sum_{i_n=1}^{p_{h,n}^K} p_{h,n}^K\mathcal{I}^2\|\phi_h(z_h^{i_n})\|_{\Lambda_{h,n}^{i_n}{}^{-1}}^2 \leq 12\sum_{n=1}^{N_h} p_{h,n}^K\mathcal{I}^2 2d_h\log(KL_\phi^2 + 1)$$
$$= 24K\mathcal{I}^2 d_h\log(KL_\phi^2 + 1). \quad (22)$$

The remaining terms can be bounded in a simpler way: by just calling $\Psi := d_h + \log(\mathcal{N}(1/\sqrt{k}, \mathcal{V}_h)) + \log(1/\delta) + \mathcal{R}_{h,n}$, we have

$$\sum_{n=1}^{N_h} \sum_{i_n=1}^{p_{h,n}^K} 12\Psi \|\phi_h(z_h^{i_n})\|_{\Lambda_{h,n}^{i_n}{}^{-1}}^2 = 12\Psi \sum_{n=1}^{N_h} \sum_{i_n=1}^{p_{h,n}^K} \|\phi_h(z_h^{i_n})\|_{\Lambda_{h,n}^{i_n}{}^{-1}}^2$$

$$\leq 12\Psi \sum_{n=1}^{N_h} 2d_h \log(KL_\phi^2 + 1)$$

$$\leq 24\Psi N_h d_h \log(KL_\phi^2 + 1),$$

where we have used Lemma 11 by [1] as before. This result, together with Equation (22) can be inserted into the previous Equation (21) to get

$$\sum_{k=1}^{K} \sum_{h=1}^{H} 2\sqrt{\alpha_{h,\rho_h(z_h^k)}^k} \|\phi_h(z_h^k)\|_{\Lambda_{h,\rho_h(z_h^k)}^k{}^{-1}} \leq \sum_{h=1}^{H} \sqrt{K}\sqrt{\sum_{k=1}^{K} 4\alpha_{h,\rho_h(z_h^k)}^k \|\phi_h(z_h^k)\|_{\Lambda_{h,\rho_h(z_h^k)}^k{}^{-1}}^2}$$

$$\leq \sqrt{24} \sum_{h=1}^{H} \sqrt{K}\sqrt{(K\mathcal{I}^2 + \Psi N_h)d_h \log(KL_\phi^2 + 1)}$$

$$\leq \sqrt{24} \sum_{h=1}^{H} (K\mathcal{I} + \sqrt{\Psi N_h K})\sqrt{d_h \log(KL_\phi^2 + 1)}.$$

Therefore, the full bound on the reget can be written, also thanks to Equation (20) as

$$R_K \leq 2\sqrt{K \log(H/\delta)} \sum_{h=1}^{H} L_\phi \sup_n \mathcal{R}_{h,n} + \sqrt{24} \sum_{h=1}^{H} (K\mathcal{I} + \sqrt{\Psi N_h K})\sqrt{d_h \log(KL_\phi^2 + 1)}.$$

Ignoring terms that are logarithmic in $K, H, L_\phi, 1/\delta$ and passing to the $\widetilde{\mathcal{O}}$ notation, we get

$$R_K \leq \widetilde{\mathcal{O}}\left(\sqrt{K} \sum_{h=1}^{H} (L_\phi \sup_n \mathcal{R}_{h,n} + \sqrt{d_h \Psi N_h}) + KH\mathcal{I}\sqrt{d_h}\right).$$

Now, note that by definition $\Psi := d_h + \log(\mathcal{N}(1/\sqrt{k}, \mathcal{V}_h)) + \log(1/\delta) + \mathcal{R}_{h,n}$, so the previous can be rewritten as

$$R_K \leq \widetilde{\mathcal{O}}\left(\sqrt{KN} \sum_{h=1}^{H} \left((L_\phi + \sqrt{d_h}) \sup_n \mathcal{R}_{h,n} + d_h + \sqrt{d_h \log(\mathcal{N}(1/\sqrt{K}, \mathcal{V}_h))}\right) + KH\mathcal{I}\sqrt{d_h}\right).$$

$\square$

Merging the previous result with Proposition 8, which bounds the size of the covering, and the fact that the good event is designed to have probability at least $1 - \delta$ (Corollary 10), we get

**Corollary 15.** *Assume that $L_\phi = \mathcal{O}(1)$ and $\sup_{h \in [H], n \in [N_h]} \mathcal{R}_{h,n} = \mathcal{O}(\sqrt{d_h})$. Then, with probability at least $1 - \delta$, CINDERELLA (Algorithm 1), with $\lambda = 1$ achieves a regret bound of order*

$$R_K = \widetilde{\mathcal{O}}\left(\sum_{h=1}^{H} N_h d_h \sqrt{K} + \sum_{h=1}^{H} \sqrt{d_h} \mathcal{I} K\right).$$

# D Computational complexity

In this section, we are going to study the computational complexity of CINDERELLA (Algorithm 1). The key computational bottleneck lies in solving the constrained continuous optimization problem presented in Equation 7. Two obstacles hinder efficient solutions to this problem.

1. In the first constrain, there is one term containing $\max_{a \in \mathcal{A}} \phi_{h+1}(s_{h+1}^\tau, a)^\top \overline{\theta}_{h+1, \rho(s_{h+1}^\tau, a)}$, which breaks linearity in $\overline{\theta}_{h+1,n}$.

2. In the third one, $\|\overline{\xi}_{h,n}\|_{\Lambda_{h,n}^k} \leq \sqrt{\alpha_{h,n}^k}$ breaks the linearity also in $\overline{\xi}_{h,n}$.

The confluence of these two challenges renders the problem computationally intractable. This is unsurprising, as even the optimization problem in ELEANOR [42], a simplified version of ours, is well-known for its computational difficulty. Fortunately, our setting allows for an inherent Bellman error, denoted by $\mathcal{I}$. This enables us to forego an exact solution to problem 7 and instead seek an approximate solution. The resulting approximation error, denoted by $\varepsilon$, can be incorporated into the Bellman error term. From theorem 2 we deduce that if $\varepsilon = \mathcal{O}(K^{-1/2})$, its dependence in the regret bound is negligible. Therefore, a viable approach might involve constructing an $\varepsilon$-grid, denoted by $\mathcal{G}$, over the variable space for $\varepsilon \approx K^{-1/2}$ and then to solve the problem with exhaustive search on $\mathcal{G}$. This idea is not completely satisfying, as the cardinality of $\mathcal{G}$ would be of order

$$|\mathcal{G}| = \mathcal{O}\left((\sqrt{K})^{3\sum_{h=1}^H N_h d_h}\right),$$

since we have three optimization variables $\widehat{\theta}_{h,n}, \overline{\xi}_{h,n}, \overline{\theta}_{h,n}$ of dimension $d_h$ for every $n \in [N_h]$. Unfortunately, at this level of generality, no strategy with polynomial computational complexity is known. As a comparison, algorithms presented in table 1 are not better. LEGENDRE-ELEANOR [25] is based on ELEANOR, so it is also intractable, GOLF [17] and NET-Q-LEARNING [34] require an optimization oracle. LEGENDRE-LSVI [25] and KOVI [41] can be implemented efficiently, but work only for simpler settings.

# E Mildly Smooth MDPs

In this section, we give our results for Mildly Smooth MDPs. First, we have to start from some notion from calculus.

## E.1 Smoothness of real functions

Let $\Omega \subset [-1, 1]^d$ and $f : \Omega \to \mathbb{R}$. We say that $f \in \mathcal{C}^\nu(\Omega)$, for $\nu \in (0, +\infty)$ if it is $\nu_*$−times continuously differentiable for $\nu_* := \lceil \nu - 1 \rceil$, and there is a constant $L_\nu(f)$ such that

$$\forall \boldsymbol{\alpha} : |\boldsymbol{\alpha}| = \nu_*, \forall x, y \in \Omega, \qquad |D^{\boldsymbol{\alpha}} f(x) - D^{\boldsymbol{\alpha}} f(y)| \leq L_\nu(f) \|x - y\|_\infty^{\nu - \nu_*},$$

where $\boldsymbol{\alpha}$ is a multi-index, i.e. a tuple of non-negative integers $(\alpha_1, \dots \alpha_d)$ and the multi-index derivative is defined as follows

$$D^{\boldsymbol{\alpha}} f := \frac{\partial^{\alpha_1 + \dots + \alpha_d}}{\partial x_1^{\alpha_1} \dots \partial x_d^{\alpha_d}}.$$

The previous set becomes a metric space when endowed with the following norm

$$\|f\|_{\mathcal{C}^\nu} := \max \left\{ \max_{|\alpha| \leq \nu_*} \|D^{\boldsymbol{\alpha}} f\|_{L^\infty}, L_\nu(f) \right\}.$$

Note that, when $\nu \in \mathbb{N}$, the previous norm simplifies as

$$\|f\|_{\mathcal{C}^\nu} = \max_{|\alpha| \leq \nu} \|D^{\boldsymbol{\alpha}} f\|_{L^\infty},$$

since the Lipschitz constant $L_\nu(f)$ of the derivatives up to order $\nu_* = \nu - 1$ corresponds exactly to the upper bound of the derivatives of order $\nu$ (which exist as a Lipschitz function is differentiable almost everywhere). For these values of $\nu$, the spaces defined here are equivalent to the spaces $\mathcal{C}^{\nu-1,1}(\Omega)$ defined in [25].

The following approximation results hold true for this function space.

**Theorem 16.** *Let us consider the Taylor polynomial $T_y^{\nu_*}[f](\cdot)$ centered in $y \in \Omega$ of order $\nu_*$. This can be written as*

$$T_y^{\nu_*}[f](x) = \sum_{\|\boldsymbol{\alpha}\|_1 \leq \nu_*} \frac{D^{\boldsymbol{\alpha}} f(y)}{\boldsymbol{\alpha}!} (x-y)^{\boldsymbol{\alpha}}.$$

*Then,*

$$\forall x \in \Omega, \ |f(x) - T_y^{\nu_*}[f](x)| \leq L_\nu(f) \|x-y\|_\infty^\nu.$$

As this formulation has the form of a scalar product over $\|\boldsymbol{\alpha}\|_1 \leq \nu_*$ of one vector with components $\frac{D^{\boldsymbol{\alpha}} f(y)}{\boldsymbol{\alpha}!}$ and another of components $(x-y)^{\boldsymbol{\alpha}}$, we rewrite it in the following form

$$T_y^{\nu_*}[f](x) = \sum_{\|\boldsymbol{\alpha}\|_1 \leq \nu_*} \frac{D^{\boldsymbol{\alpha}} f(y)}{\boldsymbol{\alpha}!} (x-y)^{\boldsymbol{\alpha}} =: \boldsymbol{w}^\top \psi_y^{\nu_*}(x). \tag{23}$$

Here, note that the length of the two vectors corresponds to $|\{\boldsymbol{\alpha} \in \mathbb{N}^d : \|\boldsymbol{\alpha}\|_1 \leq \nu_*\}| = \binom{\nu_*+d}{d} \leq \nu_*^d$. Moreover, the first only depends on $f$, while the second depends on $x$.

With all this notation settled, we can define what we call a Mildly Smooth MDP.

## E.2 Between Weak and Strong

We call Mildly Smooth MDP a process where the Bellman optimality operator outputs functions that are smooth.

**Definition 5.** *(Mildly Smooth MDP). An MDPs is* Mildly Smooth *of order $\nu$ if, for every $h \in [H]$, the Bellman optimality operator $\mathcal{T}_h$ is bounded on $L^\infty(\mathcal{Z}) \to \mathcal{C}^\nu(\mathcal{Z})$.*

Boundedness over $L^\infty(\mathcal{Z}) \to \mathcal{C}^\nu(\mathcal{Z})$ means that the operator transforms functions that are bounded $L^\infty(\mathcal{Z})$ into functions that are $\nu$-times differentiable. Moreover, there exists a constant $C_{\mathcal{T}} < +\infty$ such that $\|\mathcal{T}_h f\|_{\mathcal{C}^\nu} \leq C_{\mathcal{T}}(\|f\|_{L^\infty} + 1)$ for every $h \in [H]$ and every function $f \in \mathcal{C}^\nu(\mathcal{Z})$.

To determine how strong our assumptions are when compared to the literature, we prove the following considerations.

## E.3 Relation between the settings

$\nu-$**Kernelized** $\subset \nu-$**Mildly Smooth.** Given the definitions of the two MDP families, this point reduces to proving that the RKHS generated by the Matérn kernel of parameter $\nu > 0$ is a subspace of $\mathcal{C}^\nu(\Omega)$.

Theorem 8, part (3) from [10] ensures that, for isotropic kernels, in order to have sample path $\in \mathcal{C}^\nu(\Omega)$, a sufficient condition is for the kernel to be in $\mathcal{C}^{2\nu}(\Omega^2)$. The same theorem also shows (proof of Proposition 10) that the Matérn kernel can be written as

$$M_\nu(x,y) \propto (f_1(\|x-y\|_2^2) + \|x-y\|_2^{2\nu} f_2(\|x-y\|_2^2)),$$

where $f_1, f_2 \in \mathcal{C}^\infty(\mathbb{R})$. Note that,

$$\|x-y\|_2^2 = \sum_{i=1}^d (x_i - y_i)^2 \in \mathcal{C}^\infty(\Omega^2),$$

so that $f_1(\| \cdot - \cdot \|_2^2), f_2(\| \cdot - \cdot \|_2^2) \in \mathcal{C}^\infty(\Omega^2)$. Therefore, these two terms do not affect the overall smoothness of the kernel. This implies that the order of smoothness of $M_\nu(x,y)$ corresponds to the one of $\|x-y\|_2^{2\nu}$ which is (perhaps the most basic example of) $\mathcal{C}^{2\nu}(\Omega^2)$.

**For $\nu \in \mathbb{N}$, $\nu-$Mildly Smooth $\subset (\nu-1)-$Weakly Smooth.** This part is obvious; indeed, as for every function we have $\|f\|_{L^\infty} \le \|f\|_{\mathcal{C}^{\nu-1,1}}$ (where the last norm is defined in [25]), we have

$$\|\mathcal{T}_h f\|_{\mathcal{C}^{\nu-1,1}} = \|\mathcal{T}_h f\|_{\mathcal{C}^\nu} \le C_{\mathcal{T}}(\|f\|_{L^\infty} + 1) \le C_{\mathcal{T}}(\|f\|_{\mathcal{C}^{\nu-1,1}} + 1),$$

where in the first passage we have used the fact that for $\nu$ integer $\|\cdot\|_{\mathcal{C}^{\nu-1,1}} = \|\cdot\|_{\mathcal{C}^\nu}$ and in the second we have used the definition of $\nu-1$ Mildly-smooth MDP.

**For $\nu \in \mathbb{N}$, $(\nu-1)-$Strongly Smooth $\subset \nu-$Mildly Smooth.** Let us assume an MDP is Strongly Smooth. Indeed, for every function $f : \mathcal{S} \times \mathcal{A} \to \mathbb{R}$ we have:

$$\mathcal{T}_h f(s,a) = r_h(s,a) + \mathbb{E}_{s' \sim p_h(\cdot|s,a)}[\max_{a' \in \mathcal{A}} f(s',a')]$$

$$= r_h(s,a) + \int_{\mathcal{S}} \max_{a' \in \mathcal{A}} f(s',a') p_h(s'|s,a) \, ds'.$$

By triangular inequality, this entails:

$$\|\mathcal{T}_h f\|_{\mathcal{C}^\nu} \le \|r\|_{\mathcal{C}^\nu} + \left\| \int_{\mathcal{S}} \max_{a' \in \mathcal{A}} f(s',a') p_h(s'|\cdot) \, ds' \right\|_{\mathcal{C}^\nu},$$

where the first term $\|r\|_{\mathcal{C}^\nu} = \|r\|_{\mathcal{C}^{\nu-1,1}}$ (remember that $\nu$ is integer) is bounded by assumption and so we can focus on the second one. As all the functions involved are bounded, we can apply the theorem of exchange between integral and derivative [14] and we have, for every multi-index with $|\boldsymbol{\alpha}| \le \nu$:

$$D^{\boldsymbol{\alpha}} \int_{\mathcal{S}} \max_{a' \in \mathcal{A}} f(s',a') p_h(s'|\cdot) \, ds' = \int_{\mathcal{S}} \max_{a' \in \mathcal{A}} f(s',a') D^{\boldsymbol{\alpha}} p_h(s'|\cdot) \, ds'. \tag{24}$$

Using the abbreviation $\tilde{f}(s') = \max_{a' \in \mathcal{A}} f(s',a')$ we get:

$$\left\| \int_{\mathcal{S}} \tilde{f}(s') p_h(s'|\cdot) \, ds' \right\|_{\mathcal{C}^\nu} = \max_{|\alpha| \le \nu} \left\| D^{\boldsymbol{\alpha}} \int_{\mathcal{S}} \tilde{f}(s') p_h(s'|\cdot) \, ds' \right\|_{L^\infty}$$

$$= \max_{|\alpha| \le \nu-1} L_1 \left( D^{\boldsymbol{\alpha}} \int_{\mathcal{S}} \tilde{f}(s') p_h(s'|\cdot) \, ds' \right)$$

$$= \max_{|\alpha| \le \nu-1} \sup_{z_1, z_2 \in \mathcal{Z}} \frac{D^{\boldsymbol{\alpha}} \int_{\mathcal{S}} \tilde{f}(s') p_h(s'|z_1) \, ds' - D^{\boldsymbol{\alpha}} \int_{\mathcal{S}} \tilde{f}(s') p_h(s'|z_2) \, ds'}{\|z_1 - z_2\|_\infty}$$

$$= \max_{|\alpha| \le \nu-1} \sup_{z_1, z_2 \in \mathcal{Z}} \frac{\int_{\mathcal{S}} \tilde{f}(s')(D^{\boldsymbol{\alpha}} p_h(s'|z_1) - D^{\boldsymbol{\alpha}} p_h(s'|z_2)) \, ds'}{\|z_1 - z_2\|_\infty}$$

$$\le \sup_{z_1, z_2 \in \mathcal{Z}} \frac{\int_{\mathcal{S}} \tilde{f}(s') C_p \|z_1 - z_2\|_\infty \, ds'}{\|z_1 - z_2\|_\infty}$$

$$\le \text{Vol}(\mathcal{S}) C_p \|\tilde{f}\|_{L^\infty}$$

Where the first step is the definition of $\|\cdot\|_{\mathcal{C}^\nu}$, the second comes from the fact that, as we pointed out before, if a function $f$ is Lipschitz the $\|\cdot\|_{L^\infty}$ of its derivative corresponds to its Lipschitz constant, the third from definition of Lipschitz constant, the fourth by linearity of the derivative, the fifth one by definition of strongly $(\nu-1)-$smooth process (the part about $p_h$) and the last one by just bounding the integral with the infinity norm times the measure of the set.

**For $\nu \in \mathbb{N}$, there is an MDP that is $\nu-$Mildly Smooth but not $(\nu-1)-$Strongly Smooth.** Define an MDP where

- $\mathcal{S} = \mathcal{A} = [-1,1]$, so that $\mathcal{Z} = [-1,1]^2$.
- $r_h(z) = 0$ everywhere (any smooth function would have worked as well).

- $p_h(\cdot|s, a) = \text{Unif}(\beta s, \beta s + 1 - \beta)$ for some $\beta \in (0, 1)$.

Note that this transition function is well defined, as both $\beta s, \beta s + 1 - \beta$ are in $[-1, 1]$ but it is not even continuous, as its density corresponds to

$$p_h(s'|s, a) = \frac{\mathbf{1}_{[\beta s, \beta s + 1 - \beta]}(s')}{1 - \beta}.$$

Still, we can show that this process is Mildly smooth for $\nu = 1$. Indeed, take every $f \in L^\infty$. We have

$$\|\mathcal{T}_h^* f\|_{\mathcal{C}^1} = \max\left\{\max_{|\alpha| \leq 0} \|D^\alpha \mathcal{T}_h^* f\|_{L^\infty}, L_\nu(\mathcal{T}_h^* f)\right\}$$
$$= \max\left\{\|\mathcal{T}_h^* f\|_{L^\infty}, L_1(\mathcal{T}_h^* f)\right\}.$$

The first part $\|\mathcal{T}_h^* f\|_{L^\infty}$ is bounded by $\|f\|_{L^\infty}$ by the non-expansivity of Bellman operator. The second one corresponds to

$$L_1(\mathcal{T}_h^* f) = \sup_{z_1, z_2 \in \mathcal{Z}} \frac{\mathcal{T}_h^* f(z_1) - \mathcal{T}_h^* f(z_2)}{\|z_1 - z_2\|_\infty}.$$

We have, calling $\tilde{f}(s') = \max_{a' \in \mathcal{A}} f(s', a')$ (recall that reward is constant),

$$\sup_{z_1, z_2 \in \mathcal{Z}} \frac{\mathcal{T}_h^* f(z_1) - \mathcal{T}_h^* f(z_2)}{\|z_1 - z_2\|_\infty} = \sup_{z_1, z_2 \in \mathcal{Z}} \frac{\int_\mathcal{S} \tilde{f}(s') p_h(s'|z_1)\, ds' - \int_\mathcal{S} \tilde{f}(s') p_h(s'|z_2)\, ds'}{\|z_1 - z_2\|_\infty}$$
$$= \sup_{z_1, z_2 \in \mathcal{Z}} \frac{\int_\mathcal{S} \tilde{f}(s')(p_h(s'|z_1) - p_h(s'|z_2))\, ds'}{\|z_1 - z_2\|_\infty}.$$

We can now evaluate the numerator explicitly:

$$\int_\mathcal{S} \tilde{f}(s')(p_h(s'|s_1, a_1) - p_h(s'|s_2, a_2))\, ds' = \int_\mathcal{S} \tilde{f}(s')\left(\frac{\mathbf{1}_{[\beta s_1, \beta s_1 + 1 - \beta]}(s')}{1 - \beta} - \frac{\mathbf{1}_{[\beta s_2, \beta s_2 + 1 - \beta]}(s')}{1 - \beta}\right) ds'$$
$$= \int_\mathcal{S} \tilde{f}(s')\frac{\mathbf{1}_{[\beta s_1, \beta s_1 + 1 - \beta]}(s')}{1 - \beta}\, ds'$$
$$- \int_\mathcal{S} \tilde{f}(s')\frac{\mathbf{1}_{[\beta s_2, \beta s_2 + 1 - \beta]}(s')}{1 - \beta}\, ds'$$
$$= \frac{1}{1 - \beta}\left(\int_{\beta s_1}^{\beta s_1 + 1 - \beta} \tilde{f}(s')\, ds' - \int_{\beta s_2}^{\beta s_2 + 1 - \beta} \tilde{f}(s')\, ds'\right).$$

Now, note that the function $g(s) := \int_{\beta s}^{\beta s + 1 - \beta} \tilde{f}(s')\, ds'$ has a derivative bounded by $2\|\tilde{f}\|_{L^\infty}$ in absolute value: indeed, from the fundamental theorem of calculus,

$$|g'(s)| = |\tilde{f}(\beta s + 1 - \beta) - \tilde{f}(\beta s)| \leq 2\|\tilde{f}\|_{L^\infty},$$

so that it is $2\|\tilde{f}\|_{L^\infty}$ Lipschitz continuous. Substituting in the previous equation we get

$$\int_\mathcal{S} \tilde{f}(s')(p_h(s'|s_1, a_1) - p_h(s'|s_2, a_2))\, ds' \leq \frac{1}{1 - \beta}\left(\int_{\beta s_1}^{\beta s_1 + 1 - \beta} \tilde{f}(s')\, ds' - \int_{\beta s_2}^{\beta s_2 + 1 - \beta} \tilde{f}(s')\, ds'\right)$$
$$= \frac{1}{1 - \beta}\left(g(s_1) - g(s_2)\right) \leq \frac{2\|\tilde{f}\|_{L^\infty}}{1 - \beta}\|s_1 - s_2\|_{L^\infty}.$$

This proves that

$$\sup_{z_1,z_2\in\mathcal{Z}} \frac{\mathcal{T}_h^* f(z_1) - \mathcal{T}_h^* f(z_1)}{\|z_1 - z_2\|_\infty} = \sup_{z_1,z_2\in\mathcal{Z}} \frac{\int_\mathcal{S} \tilde{f}(s')(p_h(s'|z_1) - p_h(s'|z_2))\,ds'}{\|z_1 - z_2\|_\infty}$$

$$\leq \sup_{z_1,z_2\in\mathcal{Z}} \frac{\frac{2\|\tilde{f}\|_{L^\infty}}{1-\beta}\|s_1 - s_2\|_\infty}{\|z_1 - z_2\|_\infty}$$

$$= \frac{2\|\tilde{f}\|_{L^\infty}}{1-\beta} \leq \frac{2}{1-\beta}\|f\|_{L^\infty}.$$

Where the passage after the first inequality holds since $\|s_1 - s_2\|_\infty \leq \|z_1 - z_2\|_\infty$ (equality holds taking the supremum over $z_1, z_2$ as it is sufficient to have $a_1 = a_2$ to enforce exactly $\|s_1 - s_2\|_\infty = \|z_1 - z_2\|_\infty$) and the second is due to the fact that being $\tilde{f}(s') = \max_{a'\in\mathcal{A}} f(s',a')$ we have $\|\tilde{f}\|_{L^\infty} \leq \|f\|_{L^\infty}$. This proves that the process is $\nu-$Mildly for $\nu = 1$, while we have seen that this is not $0-$strongly smooth.

The relation between the different settings is depicted in Figure 2.

## E.4 Regret bound for Mildly smooth MDPs

As clarified in the main paper, we are going to see that for a proper choice of the partition $\mathcal{U}_h$ and of the feature map $\phi_h$, any Mildly Smooth MDP belongs to the Locally Linearizable representation class. We start with the following consideration: as $\mathcal{Z} \subset [-1,1]^d$ we can find, for all $\varepsilon$, a set $\mathcal{Z}^\varepsilon$ which forms an $\varepsilon$-cover of $\mathcal{Z}$ in infinity norm, such that $|\mathcal{Z}^\varepsilon| =: N \leq (2/\varepsilon)^d$.

Now, we define the elements of the partition and feature map in the following way. Note that, even if we could make all the elements depend on $h$, we omit this dependence as it turns out not to be necessary.

1. Now, for every $z^n \in \mathcal{Z}^\varepsilon$, we define recursively $\mathcal{Z}_n$ to be the set of points which are covered by $z^n$, formally:

   $$\mathcal{Z}_1 := \{z \in \mathcal{Z} : \|z - z^1\|_\infty \leq \varepsilon\}, \qquad \mathcal{Z}_n := \{z \in \mathcal{Z} : \|z - z^n\|_\infty \leq \varepsilon\} \setminus \cup_{\ell=1}^{n-1}\mathcal{Z}_\ell.$$

   As this sets form a partition of $\mathcal{Z}$, we can take $\mathcal{U} = \{\mathcal{Z}_n\}_{n=1}^N$.

2. Let $\rho(\cdot)$ be the function mapping each point of $\mathcal{Z}$ to the corresponding $\mathcal{Z}_n$. We define our feature map starting from equation (23) as

$$\phi(z) := \psi_{\rho(z)}^{\nu_*}(z), \tag{25}$$

   so that its length corresponds exactly to $d_{\nu_*} := \binom{\nu_*+d}{d}$.

We can prove the following fundamental result.

**Theorem 17.** *For any $f \in \mathcal{C}^\nu(\mathcal{Z})$, there are $\theta_1, \ldots \theta_n, \ldots \theta_N$, all in $\mathbb{R}^{d_{\nu_*}}$, such that*

$$\|f(\cdot) - \phi(\cdot)^\top \theta_{\rho(\cdot)}\|_{L^\infty} \leq L_\nu(f)\varepsilon^\nu.$$

*Moreover, the components of each of the vectors $\theta_n$ satisfy*

$$|\theta_n[\boldsymbol{\alpha}]| \leq \begin{cases} \|f\|_{L^\infty} & \boldsymbol{\alpha} = 0 \\ \|f\|_{\mathcal{C}^\nu} & \boldsymbol{\alpha} \neq 0. \end{cases}$$

*Proof.* Let $z \in \mathcal{Z}$ and $n = \rho(z)$. Then, if we take $\theta_n$ to be the vector with components

$$\theta_n[\boldsymbol{\alpha}] = \frac{D^{\boldsymbol{\alpha}} f(z^n)}{\boldsymbol{\alpha}!},$$

we have,

$$|f(z) - \phi(z)^\top \theta_{\rho(z)}| = |f(z) - \phi(z)^\top \theta_n|$$

$$= \left| f(z) - \sum_{\|\boldsymbol{\alpha}\|_1 \leq \nu_*} \frac{D^{\boldsymbol{\alpha}} f(z^n)}{\boldsymbol{\alpha}!} (x - z^n)^{\boldsymbol{\alpha}} \right|$$

$$= |f(z) - T_{z^n}^{\nu_*}[f](z)|,$$

where we have used both the definitions of $\phi$ and $\theta_n$. Then, using Theorem 16, we have

$$|f(z) - \phi(z)^\top \theta_{\rho(z)}| = |f(z) - T_{z^n}^{\nu_*}[f](z)|$$
$$\leq L_\nu(f) \|z - z^n\|_\infty^\nu.$$

By definition of $\mathcal{Z}^\varepsilon$, we have $\|z - z^n\|_\infty = \|z - \rho(z)\|_\infty \leq \varepsilon$, which entails the first part of the thesis. For what concerns the second one, we bound the magnitude of the vectors $\theta_n$ component by component:

$$|\theta_n[\boldsymbol{\alpha}]| = \left| \frac{D^{\boldsymbol{\alpha}} f(z^n)}{\boldsymbol{\alpha}!} \right| \leq |D^{\boldsymbol{\alpha}} f(z^n)| \leq \begin{cases} \|f\|_{L^\infty} & \boldsymbol{\alpha} = 0 \\ \|f\|_{\mathcal{C}^\nu} & \boldsymbol{\alpha} \neq 0 \end{cases},$$

where the last comes from the definition of $\|f\|_{\mathcal{C}^\nu}$ $\qquad\square$

**Theorem 18.** *Let $M$ be a Mildly Smooth MDP, and $\varepsilon \leq 1/(2C_\mathcal{T} H)^{1/\nu}$. Then, setting*

1. *$\mathcal{Z}_{h,n}$ as in Equation (8),*

2. *$\phi_h$ to be the feature map defined in Equation (25),*

*the tuple $(M, \{\mathcal{Z}_{h,n}\}_{n,h}, \{\phi_h\}_h)$ is an Locally Linearizable MDP with*

- $L_\phi = 1 + 2\sqrt{d_{\nu_*}}$.

- $\mathcal{R}_{h,n} = 2\sqrt{d_{\nu_*}} C_\mathcal{T}$.

- $\mathcal{I} \leq 2C_\mathcal{T} \varepsilon^\nu$.

*Proof.* Let us define the sets

$$\mathcal{B}_{h,n} := \left\{ \theta \in \mathbb{R}^{d_{\nu_*}} : \|\phi_h(\cdot)^\top \theta\|_{L^\infty(\mathcal{Z}_n)} \leq A_h, \|\theta\|_\infty \leq A'_h \right\}, \tag{26}$$

where $L^\infty(\mathcal{Z}_n)$ stands for the supremum norm restriceted to $\mathcal{Z}_n$. Now, let $Q_{h+1}[\boldsymbol{\theta}_{h+1}](z) = \phi(z)^\top \theta_{\rho(z)}$ with $\boldsymbol{\theta}_{h+1} \in \mathcal{B}_{h+1}$ for a set $\mathcal{B}_{h+1} = \bigtimes_{n=1}^N \mathcal{B}_{h+1,n}$. Two facts hold:

1. By the non-expansivity of the Bellman optimality operator we have
$$\|\mathcal{T}_h Q_{h+1}[\boldsymbol{\theta}_{h+1}]\|_{L^\infty} \leq \|Q[\boldsymbol{\theta}_{h+1}]\|_{L^\infty} \leq A_{h+1}. \tag{27}$$

2. By the Mildly smooth assumption (Asm. 5), we have
$$\|\mathcal{T}_h Q_{h+1}[\boldsymbol{\theta}_{h+1}]\|_{\mathcal{C}^\nu} \leq C_\mathcal{T}(\|Q_{h+1}[\boldsymbol{\theta}_{h+1}]\|_{L^\infty} + 1) \leq C_\mathcal{T}(1 + A_{h+1}). \tag{28}$$

Now, we can prove the low inherent Bellman error property (Eq. 9). To do this we apply Theorem 17 which guarantees that there are $\theta_1, \ldots \theta_n, \ldots \theta_N$, all in $\mathbb{R}^{d_{\nu_*}}$, such that

$$\|\mathcal{T}_h Q_{h+1}[\boldsymbol{\theta}_{h+1}](\cdot) - \phi_h(\cdot)^\top \theta_{\rho(\cdot)}\|_{L^\infty} \leq L_\nu(\mathcal{T}_h Q[\boldsymbol{\theta}_{h+1}]) \varepsilon^\nu.$$

Thus, if we take $\boldsymbol{\theta}_h$ as the matrix stacking all these $\theta_1, \ldots \theta_n, \ldots \theta_N$, we get that the inherent Bellman error is bounded by

$$\mathcal{I} \leq L_\nu(\mathcal{T}_h Q[\boldsymbol{\theta}_{h+1}]) \varepsilon^\nu \leq \|\mathcal{T}_h Q[\boldsymbol{\theta}_{h+1}]\|_{\mathcal{C}^\nu} \varepsilon^\nu \leq C_\mathcal{T}(1 + A_{h+1}) \varepsilon^\nu.$$

What is missing is to prove that this choice satisfies $\boldsymbol{\theta}_h \in \mathcal{B}_h$.

By triangular inequality, we have the following bound on the supremum norm

$$\|Q_h[\boldsymbol{\theta}_h](\cdot)\|_{L^\infty} = \|\mathcal{T}_h Q_{h+1}[\boldsymbol{\theta}_{h+1}](\cdot) - Q_h[\boldsymbol{\theta}_h](\cdot)\|_{L^\infty} + \|\mathcal{T}_h Q_{h+1}[\boldsymbol{\theta}_{h+1}](\cdot)\|_{L^\infty}$$
$$\leq \mathcal{I} + \|\mathcal{T}_h Q_{h+1}[\boldsymbol{\theta}_{h+1}](\cdot)\|_{L^\infty}$$
$$\leq \mathcal{I} + \|Q_{h+1}[\boldsymbol{\theta}_{h+1}](\cdot)\|_{L^\infty}$$
$$\leq C_{\mathcal{T}}(1 + A_{h+1})\varepsilon^\nu + A_{h+1}.$$

We apply again Theorem 17 having that the components of the $\theta_{h,n}$ which form $\boldsymbol{\theta}_h$ satisfy

$$|\theta_{h,n}[\boldsymbol{\alpha}]| \leq \begin{cases} \|\mathcal{T}_h Q_{h+1}[\boldsymbol{\theta}_{h+1}]\|_{L^\infty} \leq \|\mathcal{T}_h Q_{h+1}[\boldsymbol{\theta}_{h+1}]\|_{\mathcal{C}^\nu} & \boldsymbol{\alpha} = 0 \\ \|\mathcal{T}_h Q_{h+1}[\boldsymbol{\theta}_{h+1}]\|_{\mathcal{C}^\nu} & \boldsymbol{\alpha} \neq 0, \end{cases}$$

which means, by Equation (28), that

$$|\theta_{h,n}[\boldsymbol{\alpha}]| \leq C_{\mathcal{T}}(1 + A_{h+1}),$$

so that, trivially, we have $\|\theta_{h,n}\|_\infty \leq C_{\mathcal{T}}(1 + A_{h+1})$. Therefore, Equation (26) requires the condition

$$A_h \geq C_{\mathcal{T}}(1 + A_{h+1})\varepsilon^\nu + A_{h+1} \qquad A_h' \geq C_{\mathcal{T}}(1 + A_{h+1}).$$

If we set $\varepsilon \leq 1/(2C_{\mathcal{T}}H)^{1/\nu}$, the previous inequalities are satisfied if we set

$$A_h = \frac{H - h}{H} \qquad A_h' = 2C_{\mathcal{T}}.$$

Indeed, we have

$$A_h = \frac{H - h}{H}$$
$$= \frac{H - h - 1}{H} + \frac{1}{H}$$
$$= A_{h+1} + \frac{1}{H}$$
$$= A_{h+1} + \frac{2C_{\mathcal{T}}}{2C_{\mathcal{T}}H}$$
$$\geq A_{h+1} + 2C_{\mathcal{T}}\varepsilon^\nu$$
$$\geq A_{h+1} + C_{\mathcal{T}}(1 + \frac{H - h - 1}{H})\varepsilon^\nu$$
$$= A_{h+1} + C_{\mathcal{T}}(1 + A_{h+1})\varepsilon^\nu.$$

For what concerns the other term we can simply write

$$A_h' = 2C_{\mathcal{T}} \geq C_{\mathcal{T}}(1 + A_{h+1}).$$

This allows us to say that

$$\mathcal{R}_{h,n} = \text{diam}(\mathcal{B}_{h,n}) \leq \sqrt{d_{\nu_*}} A_h' = 2\sqrt{d_{\nu_*}} C_{\mathcal{T}},$$

where the presence of $\sqrt{d_{\nu_*}}$ is needed to pass from the $\infty$−norm in the definition of $\mathcal{B}_{h,n}$ to the norm two in the definition of $\text{diam}(\mathcal{B}_{h,n})$. Instead, for the feature map we have

$$L_\phi = \sup_{z \in \mathcal{Z}} \|\phi(z)\|_2$$
$$= \sup_{z \in \mathcal{Z}} \|\phi(z)\|_2$$
$$= \sup_{z \in \mathcal{Z}} \|\psi_{\rho(z)}^{\nu_*}(z)\|_2,$$

as from Definition 25. Note that, looking at the precise definition of $\psi_y^{\nu_*}(z)$ in (23), calling $\rho(z) = n$, this norm can be explicitly written as

$$\|\psi_n^{\nu_*}(z)\|_2 = \sqrt{\sum_{\|\boldsymbol{\alpha}\|_1 \leq \nu_*} (z - z^n)^{2\boldsymbol{\alpha}}}$$

$$\leq 1 + \sqrt{\sum_{1 \leq \|\boldsymbol{\alpha}\|_1 \leq \nu_*} (z - z^n)^{2\boldsymbol{\alpha}}}$$

$$\leq 1 + \sqrt{\sum_{1 \leq \|\boldsymbol{\alpha}\|_1 \leq \nu_*} 2^2} = 1 + 2\sqrt{d_{\nu_*}}.$$

Where the third passage is due to the fact that $\|z - z^n\|_\infty \leq 2$ as we have assumed $\mathcal{Z} = [-1, 1]^2$. $\quad\square$

As a consequence, merging the previous result with Theorem 14, we get the following result.

**Corollary 19.** *Under the previous assumptions, with probability at least $1 - \delta$, Algorithm 11, for $\lambda = 1$ achieves a regret bound of order*

$$R_K \leq \tilde{\mathcal{O}}\left(HC_{\mathcal{T}} d_{\nu_*} N\sqrt{K} + HC_{\mathcal{T}} d_{\nu_*}^{1/2} \varepsilon^\nu K\right).$$

Note that, with this choice we have $N \leq (2/\varepsilon)^d$ and $d = d_{\nu_*}$. Using this consideration, we can state our final regret bound, which only relies on choosing the optimal value for $N$ in the previous result. From the previous result, if $\varepsilon \leq 1/(2C_{\mathcal{T}}H)^{1/\nu}$, Algorithm 11 achieves a regret bound of order

$$R_K \leq \tilde{\mathcal{O}}\left(HC_{\mathcal{T}} d_{\nu_*} N\sqrt{K} + HC_{\mathcal{T}} d_{\nu_*}^{1/2} \varepsilon^\nu K\right).$$

Let us now set $\varepsilon = K^{-\beta}$. The order of the regret bound in $K$ corresponds to

$$R_K = \mathcal{O}(K^{1/2+\beta d} + K^{1-\beta\nu}).$$

Imposing that the two exponents are equal corresponds to setting

$$\beta d = 1/2 - \beta\nu \implies \beta = \frac{1}{2d + 2\nu}.$$

Therefore, choosing $\varepsilon = K^{-\frac{1}{2d+2\nu}}$, which corresponds to $N = \mathcal{O}(K^{\frac{d}{2d+2\nu}})$. We formalize this reasoning in our last and main theorem.

**Theorem 20.** *Let $M$ be a Mildly smooth MDP. With probability at least $1 - \delta$, Algorithm 11, for $\lambda = 1$ achieves a regret bound of order*

$$R_K \leq \tilde{\mathcal{O}}\left(Hd_{\nu_*} K^{\frac{\nu+2d}{2\nu+2d}} + H^{\frac{2\nu+2d}{\nu}}\right).$$

*Proof.* Let us set $\varepsilon = K^{-\frac{1}{2d+2\nu}}$ and, consequently, $N = \mathcal{O}(K^{\frac{d}{2d+2\nu}})$. Then, two scenarios may happen

1. If $\varepsilon \leq 1/(2C_{\mathcal{T}}H)^{1/\nu}$, Corollary 19 ensures a regret bound of order
$$R_K \leq \tilde{\mathcal{O}}\left(HC_{\mathcal{T}} d_{\nu_*} N\sqrt{K} + HC_{\mathcal{T}} d_{\nu_*}^{1/2} \varepsilon^\nu K\right) = \tilde{\mathcal{O}}\left(HC_{\mathcal{T}} d_{\nu_*} K^{\frac{\nu+2d}{2\nu+2d}}\right).$$

2. $\varepsilon > 1/(2C_{\mathcal{T}}H)^{1/\nu}$, the regret is trivially bounded by $K$. Still, as we have chosen $\varepsilon = K^{-\frac{1}{2d+2\nu}}$, the previous inequality entails $K \leq (2C_{\mathcal{T}}H)^{\frac{2\nu+2d}{\nu}}$. Therefore, the regret is bounded by the same quantity.

This completes the proof. $\quad\square$

### E.5 Discussion on $d$

A possible criticism to our main result (Theorem 4) is that the regret bound grows linearly with the feature map dimension

$$d_{\nu_*} = \binom{\nu_* + d}{\nu_*},$$

which is exponential in $d$, the dimension of $\mathcal{Z}$.

While this phenomenon may be scary, this dependence is negligible w.r.t. the one on $K$ in any learnable regime. To see this, note that, in the much easier *bandit* setting, where $H = 1$ and there is no $p_h(\cdot)$ function, [23] shows a lower bound which, for $\nu = 1$, writes as

$$R_K = \Omega\left(K^{\frac{1+d}{2+d}}\right).$$

Now, let us assume that $d \gtrsim \beta \log(K)$ for some constant $\beta > 0$. We have

$$R_K = \Omega\left(K^{\frac{1+d}{2+d}}\right) = \Omega\left(K^{\frac{1+\beta\log(K)}{2+\beta\log(K)}}\right) = \Omega\left(K^{1-\frac{1}{2+\beta\log(K)}}\right)$$
$$\geq \Omega\left(K^{1-\frac{1}{\beta\log(K)}}\right) = \Omega\left(K \cdot e^{\frac{-\log(K)}{\beta\log(K)}}\right) = \Omega\left(K \cdot e^{-1/\beta}\right) = \Omega\left(K\right).$$

This lower bound shows that our problem is not learable unless $d = o(\log(K))$. In such case, $d_{\nu_*} = \binom{\nu_*+d}{\nu_*} = o\left(\nu^d\right) = o(T^\beta)$ for every $\beta > 0$.

### E.6 Final considerations

We end the paper by making a short list of some results in the state-of-the-art for continuous MDPs that have been improved with the last theorem. All this result are valid with high probability.

- For kernelized RL with Matérn kernel, the best known regret bound is [41], which ensures
$$R_K \leq \widetilde{\mathcal{O}}\left(H^2 K^{\frac{\nu+3d/2}{2\nu+d}}\right),$$
which is super-linear for $2d > \nu$.

- For Strongly Smooth MDPs, Theorem 2 from [25] provided a regret bound of
$$R_K \leq \widetilde{\mathcal{O}}\left(H^{3/2} K^{\frac{\nu+2d}{2\nu+d}}\right),$$
which is super-linear for $d > \nu$.

- Mildly Smooth MDPs, although just defined in the present paper, are a subset of the Weakly Smooth family, for which Theorem 1 in [25] provided a regret bound that, in our notation, writes
$$R_K \leq \widetilde{\mathcal{O}}\left(C_{\text{ELE}}^H K^{\frac{\nu+3d/2}{2\nu+d}}\right).$$
This bound is not only super-linear for $2d > \nu$ but also exponential in $H$.

In all previous settings, theorem 20 ensures a regret bound for CINDERELLA of

$$R_K \leq \widetilde{\mathcal{O}}\left(H K^{\frac{\nu+2d}{2\nu+2d}}\right),$$

which is both sub-linear in any regime and polynomial in $H$. In this way, we have proved for the first time that the previous settings allow for learnable and feasible Reinforcement Learning.

### E.7 Lower bounds

A lower bound for the any of the previous settings has clearly to depend on all $K, H, \nu, d$. In particular, the regret bound is known to decrease for higher $\nu$, while it increases in the other variables. In terms of dependence on $H$, our bound is already optimal: in this setting we cannot obtain less that the $H$ dependence [42]. On the other side, the optimal order in $K$ is a major open problem in this field.

In fact, the only proved lower bound in the continuous setting involves smooth armed bandits, which can be seen as a special case of the Strongly Smooth MDPs for $H = 1$. This bound [23], of order $K^{\frac{\nu+d}{2\nu+d}}$ is matched in the bandit case (same paper). As a comparison, our bound only achieves $K^{\frac{\nu+2d}{2\nu+2d}}$, i.e. we pay a double dependence on $d$.

This poorer regret bound is not simply an artifact of the analysis, but rather a fundamental challenge that arises when moving from bandit problems to reinforcement learning (RL). In RL, the agent must estimate the value function at step $h$ using its own estimates for the value function at the next step, $h + 1$. This forces to make a covering argument on the space of candidate state-value functions (moving target problem, also a topic of the next section F), which has a detrimental effect on the regret bound.

At this point, there are essentially two possibilities:

1. The covering argument cannot be avoided, and regret order of $K^{\frac{\nu+2d}{2\nu+2d}}$ is the best we can achieve for Kernelized MDPs, Strongly Smooth MDPs and Mildly Smooth MDPs. In that case, it would be proved that continuous armed bandits are essentially easier than MDPs with continuous state-action spaces.

2. As for tabular MDPs [4], a more refined analysis allows to avoid doing the covering argument. In this way, a regret bound of order $K^{\frac{\nu+d}{2\nu+d}}$ can be proved, as the problem becomes as difficult as a smooth bandit. This would mean that, as for the comparisons multi armed bandits/tabular MDP and linear bandit/linear MDP, there is no substantial difference between continuous armed bandits and MDPs with continuous state-action spaces.

We leave this as an interesting open problem for the future advancements of the research in theoretical RL.

# F    Some observations on [38]

[38] is one of the most recent papers in the literature on RL with continuous spaces. This work, published at the celebrated Conference on Neural Information Processing Systems (2023) reported a very strong result, proving a regret bound of order $\widetilde{\mathcal{O}}(K^{\frac{\nu+d}{2\nu+d}})$ for Kernelized RL with Matérn kernel. If true, this bound would mean that it is possible to match the lower bound for the much easier case of Kernelized *bandits*.

In this section, we will pose some objections to the analysis of that paper. To this aim, we are going to refer to the updated version [39] of the same work, where the authors have corrected some minor mistakes of the published paper.

The result giving the regret bound is Theorem 2 and, precisely, equation (21) in that paper. Without digging into the algorithm that they use, we can say that, like our one, it is based on dividing $\mathcal{Z}$ into sub-regions, there called $\mathcal{Z}'$ (even if the way these regions are created is totally different). As the authors recognise in their section 4.1, the hardest point of the analysis is the so called "moving target problem": to fit the current estimate $Q_h^t$ of the state-action value function, the authors use $r_h + P_h V_{h+1}^t$, which is a random function (as $V_{h+1}^t$ is estimated as well). Therefore, standard confidence bounds do not apply straightforwardly. The way [39], as most papers, deal with this issue, is to make a covering of the space $\mathcal{V}_{h+1}$ of the state-value functions at the next step, and then make a union bound over all the possible values for $V_{h+1}^t$.

The tricky part of their analysis is that, instead of using the covering number of $\mathcal{V}_{h+1}$, they fix one region $\mathcal{Z}'$ and cover the space of functions restricted to $\mathcal{Z}'$, as it is said in the first eight lines of section 4.2. In this way, as said in the same part of the paper, the $\varepsilon-$covering number of the function class, for $\varepsilon = \mathcal{O}(\sqrt{\log(T)}/\sqrt{N_T(\mathcal{Z}')})$ (where $N_T(\mathcal{Z}')$ corresponds to the number of times region $\mathcal{Z}'$ is visited), results for be of order $\log(T)$.

Unfortunately, we find no justification for this choice. Line 7 of their algorithm, when the estimated $Q_h^t(\cdot)$ is computed, relies on equations (15-16), which do a Gaussian process regression with target $r_h(z') + V_{h+1}^t(s'_{h+1})$ for all $z' \in \mathcal{Z}'$ visited in the process. The issue here is that, while we know that $z' \in \mathcal{Z}'$, we cannot say anything about $s'_{h+1}$, which is sampled from $p_{h+1}(\cdot|z')$. The new state

is not guaranteed to belong to any specific sub-region of $\mathcal{S}$. Therefore, it is not sufficient to do a cover restricted to the value functions $\mathcal{Z}' \to [0, 1]$.

In fact, all the approaches based on value iteration suffer from this problem. For tabular MDPs, the problem of covering the space of next state value functions was well-known to prevent achieving optimal regret, and was eventually solved by [4], who found a way to avoid doing the covering at all. Note that tabular MDPs can be seen as an extreme case where $\mathcal{Z}$ is divided in regions $\mathcal{Z}'$ containing just one state-action couple, and making a cover of $\mathcal{V}_{h+1}$ (the space of candidate value functions at the next step) restricted to a single state would trivially achieve optimal regret of order $\sqrt{|\mathcal{S}||\mathcal{A}|K}$. Still, this move is known to be wrong.

