# OpenReview forum: "Local Linearity: the Key for No-regret Reinforcement Learning in Continuous MDPs"
_NeurIPS.cc/2024/Conference — NeurIPS 2024 poster_

### Official Review · Reviewer_qoL4 · 2024-07-04

**Soundness:** 3
**Presentation:** 2
**Contribution:** 2
**Rating:** 5
**Confidence:** 3

**Summary:**

The paper proposes two novel classes of MDPs in the function approximation setting:
* Mildly Smooth MDPs: in which the bellman optimality operator outputs smooth functions of degree $\nu$.
* Locally Linearizable MDPs: in which there exist a state-action feature mapping into $\mathbb R^d$ and a finite partition of the state-action space that give rise to a Q function approximator class, which is linear in the feature mapping inside each of the partitions (the vector defining the linear mapping is allowed to change between partitions).

An online algorithm (CINDERELLA) is presented and a proof of sublinear regret is given for the class of Locally Linearizable MDPs. Subsequently, it is shown that for any Mildly Smooth MDP, there exist a feature mapping and a partition (of size exponential in $d$) that constitute a Locally Linearizable MDP.
As a result, a sublinear regret bound of CINDERELLA is derived for the class of Mildly Smooth MDPs:
$\tilde O(H \nu_*^d K^{\frac{\nu + 2d}{2\nu + 2d}}$  $ + H^{\frac{2\nu + 2d}{\nu}})$,

where $\nu_* := \lceil \nu - 1 \rceil$.

**Strengths:**

* This work proposes a new class of continuous state-action MDPs in which sublinear regret is achievable
* In particular, this new class is the currently the most general one that does not suffer from lower bounds exponential in $H$

**Weaknesses:**

* The regret bound has factors that are exponential in $d$, and sublinearity in $K$ is weak (as to be expected in this level of generality).
* Similar to other algorithms in this line of work, the CINDERELLA is not computationally efficient and far from being a practical algorithm.
* I am not sure what we really learn from such a work; the technical details are complicated even at the level of exposition, and the results are very weak (as to be expected in this level of generality).

**Questions:**

- Definition 1 does not actually contain any structural assumption (for any MDP, one can pick a partitioning and feature maps if we don't require them to satisfy anything), so it doesn't make sense calling such MDPs (that have no structural properties) linearizable.
- Line 88 "(iii) we show that all families for which learnable and feasible RL is possible are included in a novel family of Mildly Smooth MDPs defined in this paper": This statement seems way to general - where did you show this?
- Does NET-Q-LEARNING indeed obtain regret bound that is independent of $\nu$ for the Strongly Smooth MDP setting? In this case is it strictly better than CINDERELLA for this MDP class?

---

> ### Author Rebuttal · Authors · 2024-08-06
>
> Q: _The regret bound has factors that are exponential in $d$._
>
> Lower bounds show that exponential dependence is $d$ cannot be avoided. If we look at Gaussian bandits with the squared-exponential kernel, a problem family that is strictly contained also in the Strongly Smooth class, [1] ensures an $\exp(d)$ regret lower bound (see Table 1 of [1]). Therefore, exponential scaling in $d$ should not be considered a limitation of our work: it is well-known that RL problems on continuous state-action space can be really hard even for moderately sized $d$.
>
> If, in linear MDP problems, it is well known that $d$ may be very large, almost of the order of $K$, the same thing does not apply here. In fact, the lower bound, even for the simple case $\nu=1,H=1$, entails that $d$ is much smaller than $\log(K)$:
> $$R_K=\Omega(K^{\frac{d+1}{d+2}})\quad \text{and } \quad  d\approx c\log(K) \implies R_K=\Omega(K^{\frac{c\log(K)+1}{c\log(K)+2}})=\Omega(K).$$
> This means that, unless $d\ll \log(K)$, _the problem is provably unlearnable_.
>
> Q: _sublinearity in $K$ is weak_
>
> **About $K$, we do not only have sub-linearity**: of large importance is that for $\nu\to \infty$ one recovers regret of $\sqrt K$ as for tabular or linear MDPs. In fact, having $\nu\rightarrow +\infty$ is very common in physical scenarios: the presence of a Gaussian noise in the transition function is sufficient to guarantee this condition. On the other side "only sublinearity", meant as having a regret order that is slightly less than one, is what competitor works get (See Net-Q-Learning).
>
> Q: _I am not sure what we really learn from such a work; the technical details are complicated even at the level of exposition._
>
> The idea of using a feature map that splits across a finite number of regions of the state space is rather intuitive, in our opinion.
>
> Before this work, there were mainly two approaches to continuous MDPs. 1) Discretization-based (e.g. NET-Q_LEARNING) algorithms cannot fully exploit smoothness: the regret scales as $K^{\frac{1+d}{2+d}}$ even for $\nu\gg 1$, which makes them suboptimal. 2) Linearization-based approaches [2], which are able to exploit the parameter $\nu$, achieving regret $\sqrt K$ for $\nu \to \infty$, but fail to give a sub-linear regret bound for $d>2\nu$. The reason can be found in a well-known problem of linear models with misspecification [3].
>
> Q: _Definition 1 does not actually contain any structural assumption (...), so it doesn't make sense calling such MDPs (that have no structural properties) linearizable._
>
> As the reviewer correctly states, Definition 1 does not contain any structural assumptions. In fact, as stated in the paper, potentially, every MDP belongs to this class.
> The point is that the guarantees are "non-trivial" only if the approximation error $\mathcal I$, which is built based on definition 1, is small. Having definition 1 + small $\mathcal I$ corresponds to the intuitive idea of locally linearizable MDP.
> We will modify the presentation to just define $\mathcal I$ and then call, heuristically, Locally Linearizable MDPs, the ones for which this quantity is small, similarly to what was done in [4].
>
> Our novelty lies in **combining the two approaches by** splitting the parameters of the feature map (linearization) across different regions (discretization). In this way, we bypass the problem in [1], while having the same benefits of linearization approaches.
>
> This technique partially solves a long-standing open problem in the Kernelized MDP community (see COLT 2024 Open Problem: Order Optimal Regret Bounds for Kernel-Based Reinforcement Learning), as kernelized MDPs are a subclass of our setting.
>
> Q: _Line 88 "(iii) we show that all families for which learnable and feasible RL is possible are included in a novel family of Mildly Smooth MDPs defined in this paper": This statement seems way to general - where did you show this?_
>
> As we will clarify in the final version, this sentence does not mean to say "all families that exist" but rather "all families that were defined in the continuous RL literature, to the best of our knowledge". This sentence draws from some results in [2] saying that all the parametric families (LQRs, Linear MDPs, Kernelized MDPs, etc.) with a feasible regret bound are contained in the Strongly Smooth class, which is itself contained in the Mildly Smooth one.
>
> Q: _Does NET-Q-LEARNING indeed obtain regret bound that is independent of_
> $\nu$ _for the Strongly Smooth MDP setting? In this case is it strictly better than CINDERELLA for this MDP class?_
>
> We remark that, as the lower bound decreases with $\nu$, having a regret guarantee which improves for higher $\nu$ is a desirable feature. Yes, the regret of NET-Q-LEARNING is independent of $\nu$ (for $\nu\ge 1$), but this is a massive drawback w.r.t. our algorithm.
>
> In fact, if one ignores the smoothness of the process, no better regret bound can be obtained than $K^{\frac{d+1}{d+2}}$, the one of NET-Q-LEARNING. But ignoring all the smoothness of the process reveals suboptimal: the higher the $\nu$, the lower the regret. Algorithms like ours, or the ones proposed in [2], as well as the ones coming from the Kernelized RL literature, are able to approach regret order $\sqrt K$ for $\nu \to \infty$, while NET-Q-LEARNING on MDPs cannot do better than $K^{3/4}$ (as $d\ge 2$).
>
> Furthermore, note that many real settings are characterized by a very high value for $\nu$: if the transition of the MDP is affected by a Gaussian noise, $\nu=+\infty$. Ideed, convolution of any function with a Gaussian density becomes infinite times differentiable.
>
> [1] Vakili et al., On Information Gain and Regret Bounds in Gaussian Process Bandits.
>
> [2] Maran et al., No-Regret Reinforcement Learning in Smooth MDPs.
>
> [3] Lattimore et al., Learning with Good Feature Representations in Bandits and in RL with a Generative Model.
>
> [4] Zanette et al., Learning Near Optimal Policies with Low Inherent Bellman Error

---

> > ### Comment · Area_Chair_s3k2 · 2024-08-13
> > **Reminder to address the author's rebuttal**
> >
> > Dear qoL4, please try to address the author's response and elaborate if it changed your opinion and score.
> >
> > Thank you.

---

> > ### Comment · Reviewer_qoL4 · 2024-08-13
> > **Reply**
> >
> > Thank you for the detailed rebuttal.
> >
> > To be clear, I do not contend the results in the paper did not require technical novelty.
> > In addition, my complaints about the inefficiency and weak guarantees were not to say you could have improved them.
> >
> > My concerns are all with regards to the fact that the problem studied involves many technical details adding a layer of complication over a setting which is already quite niche, one that fundamentally leads to inefficient algorithms and weak guarantees.
> >
> > Hence, I will maintain my score (but certainly do not oppose acceptance).

---

### Official Review · Reviewer_TQwD · 2024-07-10

**Soundness:** 4
**Presentation:** 3
**Contribution:** 2
**Rating:** 6
**Confidence:** 4

**Summary:**

This paper introduces a notion of local linearization, which is then applied to smooth MDPs. It generalizes the "Eleanor" algorithm into "Cinderella," which, by avoiding a "cardinality of N" term on the suboptimality with respect to the inherent Bellman error, gets sublinear regret for all classes of smooth problems.

EDIT: I appreciate the authors' detailed feedback. My score remains unchanged.

**Strengths:**

This paper is written very clearly, exposes all the strengths and weaknesses of its contributions, and makes connections to prior work in a thorough and honest fashion. The paper explain, in particular, why Cinderella is inadequate, and how the parameter learning is decoupled in a way to permit efficient learning. This requires additional technical subtlety in the analysis, but with it comes a bound which would be intractable otherwise.

**Weaknesses:**

This paper is well executed and technically solid, but I believe its main weakness is that the results are not particular surprising to an expert in the area. It seems clear that one can decompose "smooth" MDPs into local linear ones (this has been done, e.g. with zooming bandits and zooming MDPs), and is rather unsurprising that a careful way of handling the analysis makes these objects learnable. I don't seem to see any particular new techniques or insights, and so it is hard for me to be incredibly excited about the result. Still, the paper is executed commendably and for that I lean towards acceptance.

There are also some limitations with presentation. Chief among them, the authors should do more to emphasize the algorithmic differences between Eleanor and Cinderalla (as they did so well with Eleanor's limitations for regret). The extent of the discussion seems limited to the sentence "Difference with ELEANOR stays in the fact that parameters relative to different regions are learned separately", which (a) bears elaboration, (b) is somewhat hidden in the mass of text, and (c) is not grammatically correct. I would encourate the authors to explain what techniques and ideas are needed to update the algorithm and analysis for the resultant guarantees.

**Questions:**

What, if any, are the new analysis techniques developed for this setting?

**Limitations:**

Novelty, somewhat incremental. Moreover, this paper has the computational inefficiency challenges associated with many algorithms in the field, and, like any non-parametric style method, solves exp(dimension, horizon). Lastly, there do not seem to be any lower bounds that characterize the correct exponents.

---

> ### Author Rebuttal · Authors · 2024-08-06
>
> Q: _The extent of the discussion seems limited to the sentence "Difference with ELEANOR stays in the fact that parameters relative to different regions are learned separately" ... What, if any, are the new analysis techniques developed for this setting?_
>
> We thank the reviewer for giving us the opportunity to clarify this point. Section 3.1 does not explain the novelty of our approach, which is somehow hidden in the appendix; we are going to explain it here and add it to the final version.
>
> Before this work, there were two main approaches to continuous MDPs.
> 1. Discretization-based (e.g., the Zooming algorithm mentioned by the reviewer) algorithm cannot fully exploit smoothness: due to the "zero-order approximation" the regret scales as $K^{\frac{1+d}{2+d}}$ even for $\nu\gg 1$, which makes them suboptimal.
> 2. Linearization-based approaches [3], which are able to exploit the parameter $\nu$, achieving regret $\sqrt K$ for $\nu \to \infty$, but fail to give a sub-linear regret bound for $d>2\nu$. The reason can be found in a well-known problem of linear models with misspecification [1].
>
> Our novelty lies in **combining the two approaches by** splitting the parameters of the feature map (linearization) across different regions (discretization). In this way, we are able to get the best of both worlds: using our local linear approximation, we move from the zeroth-order approximation by Zooming/Net-Q-Learning to an arbitrary order $\nu$, which gives sharper bounds for smooth functions. At the same time, the locality of the feature map bypasses the problem in [1], since it allows us to avoid using tools from linear algebra (like "optimal design" argument) [1,2], which has brought the same regret as [3].
> This feature of _Cinderella_ shows that the former techniques can be replaced with the "pigeonhole argument", widely employed in the field of tabular MDPs. As a result, we get the first regret bound for the setting that achieves $\sqrt K$ for $\nu \to \infty$ while being always sub-linear.
>
> In particular, we think that bridging these two approaches - which do not seem to have anything in common - and finding a strategy to elude the lower bound presented in [1] are relevant technical contributions to the learning theory community.
>
> Q: _this paper has the computational inefficiency challenges associated with many algorithms in the field, and, like any non-parametric style method, solves exp(dimension, horizon)._
>
> As the reviewer correctly states, the same computational challenges exists also for celebrated algorithms which work at a similar level of generality. We remark that [4, 5] require access to an optimization oracle over the space of candidate $Q$ functions. This means that if one is interested in approximating a $C^\nu$ function, as in our case, an optimization over a space of $O(2^{(1/\varepsilon)^d})$ functions is needed (if one is satisfied by $\varepsilon-$cover, otherwise $+\infty$).
>
> Q: _Lastly, there do not seem to be any lower bounds that characterize the correct exponents._
>
> As we write in Appendix E.7 Lower bounds, matching the lower bound for this setting is a challenging open problem. The best-known lower bound for the order in $K$ is $\frac{\nu+d}{2\nu+d}$, which is inherited from smooth bandits. Nonetheless, even achieving it for Kernelized MDPs with Matern Kernel - a problem which is a subfamily of Strongly Smooth MDPs, which are in turn a subfamily of our setting - is an open problem (see COLT 2024 Open Problem: Order Optimal Regret Bounds for Kernel-Based Reinforcement Learning). Note that, for the former class of processes, a regret order of $\frac{\nu+2d}{2\nu+2d}$ was never proved before this paper; therefore, our result is novel even when restricted to the family of Kernelized MDPs.
>
>
> [1] Lattimore et al., Learning with Good Feature Representations in Bandits and in RL with a Generative Model
>
> [2] Zanette et al., Learning Near Optimal Policies with Low Inherent Bellman Error
>
> [3] Maran et al., No-Regret Reinforcement Learning in Smooth MDPs
>
> [4] Jin et al. "Bellman eluder dimension: New rich classes of rl problems, and sample-efficient algorithms."
>
> [5] Du et al. "Bilinear classes: A structural framework for provable generalization in rl."

---

> > ### Comment · Reviewer_TQwD · 2024-08-13
> > **Maintaining my score**
> >
> > I thank the authors for their detailed feedback. I still don't feel that the novelty warrants a score increasing, but I do appreciate the authors' clarification.

---

### Official Review · Reviewer_1ViU · 2024-07-12

**Soundness:** 3
**Presentation:** 3
**Contribution:** 2
**Rating:** 5
**Confidence:** 3

**Summary:**

The paper introduces the concept of Locally Linearizable MDPs, a class of MDPs that generalizes existing ones like Linear MDPs and MDPs with low inherent Bellman error. In this model, the state-action space is partitioned into $N$ regions, where the Q-functions belong to a class that allows the result of the Bellman optimality operator to be well approximated by a linear function within each region, up to some "Inherent Bellman Error." The authors propose CINDERELLA, a no-regret (computationally inefficient) algorithm designed for this class, achieving regret bound of $Nd\sqrt{K} + \sqrt{d}\mathcal{I}K$, where $N$ is the number of partitions, $d$ is the dimension of the state-action feature space, $K$ is the number of episodes and $\mathcal{I}$ is the Inherent Bellmann Error.

Next, they present the class of "Mildly Smooth MDPs". This class is similar yet slightly less general than "Weakly Smooth MDPs," which assume that the output of the Bellman operator on a smooth function remains smooth. Still, Mildly Smooth MDPs are more general than "Strongly Smooth MDPs" and thus positioned somewhere between the two. The authors show that Mildly Smooth MDPs are Locally Linearizable for some partition and Inherent Bellmann Error and by that achieve regret of $H v^d K^{\frac{\nu+2d}{2\nu+2d}} + H^{\frac{2\nu + 2d}{\nu}}$ where $\nu$ is the smoothness parameter of the class.

**Strengths:**

The paper is well-crafted and clearly positions itself within the existing literature. It employs a well-structured pedagogical approach. It first presents a straightforward solution for Locally Linearized MDPs and proceeds to point out that this naive approach's regret bounds are insufficient for deducing a non-trivial regret bound for Mildly Smooth MDPs. This leads to the introduction of their algorithm, which offers improved regret bounds and leads to their main result.

The authors conduct a comprehensive analysis of the regret bounds, discussing each component to assess its necessity and provide a comparison with prior work. For the most part, the paper is transparent and addresses its limitations.

**Weaknesses:**

The nature and definition of Mildly Smooth MDPs are not very natural, and it is hard to agree that learning becomes substantially more feasible compared to Weakly smooth MDPs. Specifically, the authors mention that in Weakly Smooth MDPs the regret bound must scale exponentially with $H$, which they consider "statistically unfeasible". However the proposed approach results in a regret bound that scales exponentially with $d$, the dimension of the feature space. This raises similar concerns about the practical feasibility of the setting. The authors claim that for $\nu \to \infty$ the regret becomes of order $\sqrt{K}$, but this is not clearly derived from the provided regret bounds, leaving it unclear whether the result generalizes existing results in Strongly Smooth MDPs.

**Questions:**

* Does [24] also have similar dependency in $d_{v_*}$? If not, I wouldn't consider your regret as *"improved regret bounds for Strongly Smooth MDPs"*
* Could you clarify how you achieve $\sqrt{K}$ regret in the limit $\nu \to \infty$?

**Limitations:**

For the most part, the paper addresses its limitations.

---

> ### Author Rebuttal · Authors · 2024-08-06
>
> Q: _However the proposed approach results in a regret bound that scales exponentially with
> , the dimension of the feature space. This raises similar concerns about the practical feasibility of the setting. (...) Does [24] also have similar dependency in_
> $d_{\nu_*}$? _If not, I wouldn't consider your regret as "improved regret bounds for Strongly Smooth MDPs"_
>
> Yes, also [24] enjoys a regret that is exponential in $d$. This can be seen either from the proof of Theorem 11, where the regret scales as $\sqrt{Vol(S\times A)}=2^{d/2}$ or from the fact that the length of their feature map is $\binom{N+d}{N}\approx N^d$ (see last paragraph before section 4.3).
> In fact, this exponential dependence **cannot be avoided**. Indeed, if we look at Gaussian bandits with the squared-exponential kernel, a problem family that is strictly contained also in the Strongly Smooth class, [1] ensures an $\exp(d)$ regret lower bound (see Table 1 of [1]).
>
> Regarding the comparison between our regret bound and the one of [24], we have to remark that not only our regret order is always strictly better, as
>
> $$\frac{\nu+3/2d}{2\nu+d}>\frac{\nu+2d}{2\nu+2d}\qquad \forall d,\nu>0,$$
>
> but also, and most importantly, our result is always sub-linear, while [24] gets vacuous for $d>2\nu$. Therefore, our improvement over the regret bounds is not limited to avoiding the exponential dependence in $H$.
>
> Q: _Could you clarify how you achieve regret_ $\sqrt K$ _in the limit_ $\nu\to \infty$?
>
> The regret order, for $\nu\to +\infty$, gets
> $$\lim_{\nu\to \infty}\frac{\nu+2d}{2\nu+2d}=1/2.$$
>
> In general, this is the condition that papers on continuous space / RKHS seek, as it shows that whenever functions tend to be infinitely smooth the optimal order is recovered.
>
> [1] Vakili et al., On Information Gain and Regret Bounds in Gaussian Process Bandits.

---

> > ### Comment · Reviewer_1ViU · 2024-08-12
> >
> > The first term in your regret bound scales as $Hd_{v^{*}}K^{\frac{v+2d}{2v+2d}}$.
> >
> > $\lim_{v \to \infty} d_{v^\*} = \infty$, so how do you achieve regret $\sqrt{K}$ in the limit $v\to\infty$?

---

> > > ### Author Response · Authors · 2024-08-13
> > > **On regret of $O(\sqrt K)$**
> > >
> > > As the reviewer correctly points, the constant in the bound diverges as $\nu \to +\infty$.
> > >
> > > To overcome this issue, we note that, since the smooth spaces satisfy
> > >
> > > $$\nu_1<\nu_2\implies C^{\nu_2}\subset C^{\nu_1},$$
> > >
> > > we can limit the magnitude of $\nu$ in case it is much bigger than $K$, and apply the regret bound for a lower value $\nu'$. Specifically, if $\nu=+\infty$, we can apply the bound for $\nu':=\log(K)<\nu$, which results in
> > >
> > > $$R_K\le \widetilde{\mathcal O}(d_{\nu'}K^{\frac{\nu'+2d}{2\nu'+2d}})\le \widetilde{\mathcal O}(\log(K)^dK^{1/2+\frac{d}{d+\log(K)}})\to \widetilde{\mathcal O}(K^{1/2}),$$
> > >
> > > where we have used line 253 to bound $d_{\nu'}$. The appearance of $\log(K)^d$ should not scare: this term is necessary even  in the simpler case of bandits with SE kernel, as the lower bound in [1] shows.
> > >
> > > Before the reviewer's comment, we only considered $1/2$ to be the limit of the order, i.e. the exponent of $K$, as done in most of the works on the field. Still, we think that the result of this discussion is valuable, and we are going to add it to the final version of the papaer.
> > >
> > >
> > > [1] Vakili et al., On Information Gain and Regret Bounds in Gaussian Process Bandits.

---

### Official Review · Reviewer_MjfW · 2024-07-15

**Soundness:** 3
**Presentation:** 4
**Contribution:** 3
**Rating:** 7
**Confidence:** 2

**Summary:**

The paper discusses the concept of local-linear MDPs which is a general representation class of MDPs that extends previous works on learnable (sublinear regret in $K$) and feasible (polynomial regret in $H$) episodic MDPs.

**Strengths:**

The paper considers important complexity questions associated with the continuous episodic MDP, and is very well-written. Compared to the much more well-studied tabular episodic case, the complexity of continuous episodic MDPs is much more challenging to characterize. The authors identified a class of mildly smooth MDPs and show one can achieve no-regret learning in this regime. The regret bound still has an exponential dependence in $H$ but the authors demonstrated through a continuous bandit special case this exponential dependence is unavoidable. The class of mildly smooth MDPs is larger than the previously studied classes such as kernelized MDPs and strongly smooth MDPs where no-regret learning is possible.

**Weaknesses:**

Even though the theoretical development is sound and complete, I think the authors could do better in further highlighting their contributions which I will use the questions below to elaborate.

**Questions:**

Can you give some concrete examples (can be purely constructed examples or from real-world applications) of mildly smooth MDPs that are not strongly smooth? Since I'm not very familiar with the literature, this could help me further understand what are the additional new problem instances you are able to characterize no-regret learning.

Moreover, can you discuss what the current gap between your $\tilde{O}(Hd_{\nu_*}K^{\frac{\nu+2d}{2\nu+2d}}+H^{\frac{2\nu+2d}{\nu}})$ is, especially in terms of $K$?

**Limitations:**

There is no obvious limitation in this paper.

---

> ### Author Rebuttal · Authors · 2024-08-06
>
> We point out that the regret of our algorithm does _not_ depend exponentially in $H$. Exponential dependence cannot be avoided for the Lipschitz class, and this work is also born to find a class of MDPs that is able to elude this undesirable phenomenon.
>
> Q: _Can you give some concrete examples (can be purely constructed examples or from real-world applications) of mildly smooth MDPs that are not strongly smooth?_
>
> To have an idea of an MDP that is Midly Smooth but not Strongly Smooth, one can think about transition functions of the form
>
> $$s'=f(s,a)+U,\qquad U\sim \text{Unif}(b_1,b_2).$$
>
> Indeed, the density function of the uniform noise is _not continuous_, which makes this class of MDPs not Strongly Smooth. Still, we argue that if $f(\cdot,\cdot)$ is not itself discontinuous, this class of MDPs is Mildly smooth. For more details, see the example at the end of page 29 and the whole page 30. Of course, this is true not only for uniform noise, but also for any noise admitting a bounded density function, even discontinuous. We argue that this family of processes is rather realistic and, therefore, the generalization has practical relevance.
>
> Q: _Moreover, can you discuss what the current gap between your_ $\widetilde O(Hd_{\nu_*}K^{\frac{\nu+2d}{2\nu+2d}}+H^{\frac{2\nu+2d}{\nu}})$ _is, especially in terms of_ $K$?
>
> We do not understand which "gap" the reviewer was interested in, so we kindly ask the reviewer to clarify it.

---

> > ### Comment · Reviewer_MjfW · 2024-08-13
> >
> > Thanks for the clarifications. By "gap"  I mean the gap between the current bound and the information-theoretic lower bound on the regret.

---

> > > ### Author Response · Authors · 2024-08-13
> > > **Official comment by authors**
> > >
> > > We thank the reviewer for this question, which allows us to highligh some future goals of this line of research on MDP with continuous state and action space.
> > >
> > > Matching the lower bound for this setting is a challenging open problem. The best-known lower bound for the order in
> > > $K$ is $\frac{\nu+d}{2\nu+d}$, which is inherited from smooth bandits. Nonetheless, even achieving it for Kernelized MDPs with Matern Kernel - a problem which is a subfamily of Strongly Smooth MDPs, which are in turn a subfamily of our setting - is an open problem (see COLT 2024 Open Problem: Order Optimal Regret Bounds for Kernel-Based Reinforcement Learning). Note that, for the former class of processes, a regret order of $\frac{\nu+2d}{2\nu+2d}$ was never proved before this paper; therefore, our result is novel even when restricted to the family of Kernelized MDPs.
> > >
> > > Why is this problem so hard? The root of the issue is to be found in the so-called "covering arguments", which regularly appear in the analysis of regret for value-based methods; for an example see Lemma D.6 from the seminal paper [1] on Linear MDPs.
> > > Indeed, as a value-based algorithm needs to estimate the value function at each step $h$ from the same function at step $h+1$, the corresponding regression problem has a stochastic target. When trying to solve a regression problem of this type, where the random target is *not* independent from the samples used for estimation, we pay a penalization term which depends on the covering number of the function class.
> > >
> > > Unfortunately, the function class used in our problem - or in Kernelized MDPs - is very big, and this has a pernicious effect on the final regret bound. In previous papers, this led to also to superlinear regret bounds (Table 1), while here, with the idea of local linearity, we are able to get $K^{\frac{\nu+2d}{2\nu+2d}}$. Doing better than this would mean finding a way to avoid the covering argument, but we do not know if this is possible: again, the lower bound is proved for Continuous Armed Bandits, where covering arguments are not needed.
> > >
> > > [1] Chi Jin, Zhuoran Yang, Zhaoran Wang, and Michael I Jordan. Provably efficient reinforcement
> > > learning with linear function approximation. In Conference on Learning Theory,

---

> > > > ### Comment · Reviewer_MjfW · 2024-08-13
> > > >
> > > > Thank you for the answer!

---

### Official Review · Reviewer_RTnn · 2024-07-22

**Soundness:** 3
**Presentation:** 3
**Contribution:** 3
**Rating:** 6
**Confidence:** 3

**Summary:**

This paper studied a structural property called local linearity that makes continuous MDPs learnable. In particular, local linearity means that the continuous state-action space can be partitioned into multiple regions and in each region, the Q-function is linear w.r.t. a unique (different) parameter. The paper first proposed an algorithm to learn MDPs with local linearity and proved that the regret is sublinear. Then, the paper showed that mildly smooth MDPs, where the Bellman operator is smooth, are locally linearizable. So that it demonstrated the broadness of local linearity.

**Strengths:**

1. The studied problem is important.
2. The new concept of local linearity is interesting and indeed capture a broad class of MDPs according the paper.

**Weaknesses:**

1. The proposed algorithm is computationally inefficient. It needs to run over all regions while the number of regions might be exponentially large according to section 4.

**Questions:**

Please see the weaknesses part.

**Limitations:**

No further limitations need to be addressed.

---

> ### Author Rebuttal · Authors · 2024-08-06
>
> Q: _The proposed algorithm is computationally inefficient. It needs to run over all regions while the number of regions might be exponentially large according to section 4._
>
> It is true that the number of regions is exponential in $d$. This makes the computational complexity exponential in this parameter. Still, we argue that this is not a downside w.r.t. the state of the art.
>
>  1. State of the art: other algorithms for other very general continuous RL settings require access to an optimization oracle [1, 2] over the space of candidate $Q$ functions. This means that if one is interested in approximating a $C^\nu$ function, as in our case, an optimization over a space of $O(2^{(1/\varepsilon)^d})$ functions is needed (if one is satisfied by $\varepsilon-$cover, otherwise $+\infty$).
>
>  2. Magnitude of $d$: if, in linear MDP problems, it is well known that $d$ may be very large, almost of the order of $K$, the same thing does not apply here. In fact, the lower bound, even for the simple case $\nu=1,H=1$, entails that $d$ is much smaller than $\log(K)$:
>     $$R_K=\Omega(K^{\frac{d+1}{d+2}})\quad \text{and } \quad  d\approx c\log(K) \implies R_K=\Omega(K^{\frac{c\log(K)+1}{c\log(K)+2}})=\Omega(K).$$
>     This means that, unless $d\ll \log(K)$, _the problem is provably unlearnable_.
>     Therefore, an exponential dependence in $d$ is not as bad as it may seem.
>
>
> [1] Jin et al. "Bellman eluder dimension: New rich classes of rl problems, and sample-efficient algorithms."
>
> [2] Du et al. "Bilinear classes: A structural framework for provable generalization in rl."

---

> > ### Comment · Reviewer_RTnn · 2024-08-13
> >
> > Thanks for the response. The second point seems reasonable. I would raise my score to 6.

---

### Decision · Program_Chairs · 2024-09-25

**Decision:**

Accept (poster)

**Comment:**

This paper investigates regret minimization in continuous and smooth MDPs. The authors first study a linear MDP setting where the model parameter (\theta) may change across different partitions of the state space, and the feature map (\phi) is fixed throughout the space. They design a new algorithm, Cinderella, which generalizes an existing algorithm for the linear MDP setting (Eleanor). The authors then apply their generalization to the continuous and smooth MDP setting by defining the feature space as polynomials, leveraging Taylor's expansion for smooth functions. Their results improve upon prior works for the Mildly Smooth MDP setting (Definition 3, which assumes Bellman operator smoothness). Additionally, they derive regret bounds for Cinderella in the Strongly Smooth and Kernel MDP settings.

As pointed out by the reviewers, this work has some limitations:
1. Although the paper is well-executed, it is unclear which new analysis tools the authors introduce and how other researchers can utilize these tools to enhance their results or develop new algorithms based on these findings.
2. Cinderella lacks computational efficiency, which is a common issue with optimistic algorithms.

Despite these limitations, I believe this paper makes a significant contribution by formalizing an intuitive idea and advancing our understanding of RL algorithms for continuous MDPs, hence I recommend accepting this paper. I encourage the authors to incorporate the reviewers' comments in the final draft and include the analysis mentioned in their discussion with Reviewer 1ViU.